# KSHV episomes reveal dynamic chromatin loop formation with domain-specific gene regulation

Mel Campbell[1], Tadashi Watanabe [2], Kazushi Nakano [1], Ryan R. Davis [3], Yuanzhi Lyu [1], Clifford G. Tepper [4], Blythe Durbin-Johnson[5], Masahiro Fujimuro[2] & Yoshihiro Izumiya [1,4,6]

The three-dimensional structure of chromatin organized by genomic loops facilitates RNA polymerase II access to distal promoters. The Kaposi's sarcoma-associated herpesvirus (KSHV) lytic transcriptional program is initiated by a single viral transactivator, K-Rta. Here we report the KSHV genomic structure and its relationship with K-Rta recruitment sites using Capture Hi–C analyses. High-resolution 3D viral genomic maps identify a number of direct physical, long-range, and dynamic genomic interactions. Mutant KSHV chromosomes harboring point mutations in the K-Rta responsive elements (RE) significantly attenuate not only the directly proximate downstream gene, but also distal gene expression in a domain-specific manner. Genomic loops increase in the presence of K-Rta, while abrogation of K-Rta binding impairs the formation of inducible genomic loops, decreases the expression of genes networked through the looping, and diminishes KSHV replication. Our study demonstrates that genomic architectural dynamics plays an essential role in herpesvirus gene expression.

[1] Department of Dermatology, School of Medicine, University of California Davis (UC Davis), Sacramento, CA 95817, USA. [2] Department of Cell Biology, Kyoto Pharmaceutical University, Kyoto 607-8412, Japan. [3] Department of Pathology and Laboratory Medicine, School of Medicine, UC Davis, Sacramento, CA 95817, USA. [4] Department of Biochemistry and Molecular Medicine, UC Davis School of Medicine, Sacramento, CA 95817, USA. [5] Division of Biostatistics, Department of Public Health Sciences, UC Davis School of Medicine, Sacramento, CA 95817, USA. [6] UC Davis Comprehensive Cancer Center, Sacramento, CA 95817, USA. Mel Campbell, Tadashi Watanabe, and Kazushi Nakano contributed equally to this work. Correspondence and requests for materials should be addressed to Y.I. (email: yizumiya@ucdavis.edu)

Tissue-specific cellular gene expression is regulated by the formation of active chromatin hubs (ACHs) at enhancer regions of the genome, where many tissue specific-gene promoters are brought into proximity[1]. As reviewed by Palstra et al.[2], the concept of ACHs originated, in part, from the fact that the protein concentration of many nuclear factors is below the dissociation constant of protein-protein or protein-DNA interactions. Accordingly, it is necessary to have mechanisms to increase the local concentration of nuclear factors at a given chromatin site. Transcription factors pinpoint their binding sites by three-dimensional scrutiny of nuclear space, and the formation of productive transcription complexes on DNA is intrinsically dynamic[3,4]. A higher concentration of factors favors efficient binding to DNA templates by facilitating rapid re-association of dissociating factors at the same or abutting sequences. Thus, the concentration of transcription factors and co-factors near transcription initiation sites is a sensitive limiting component determining the number of transcripts produced. Therefore, spatial and temporal clustering of cognate binding sites is proposed to be an important means to boost the local concentration of factors and thus is indispensable for the regulation of the transcriptional rate of genes[5].

Development of chromosome conformation capture (3C) techniques has permitted the examination of ACH formation, and numerous studies have indeed demonstrated widespread occurrence of stimulus-responsive enhancer-promoter and promoter-promoter interactions between co-regulated genes[6–8]. It is important to note that core promoters typically only support low-level basal transcription; cis-regulatory modules or enhancers actually carry most of the regulatory information for gene expression[9]. These elements can be located near the core promoter or distal to a target gene which is typically found within accessible chromatin[10,11]. The combined input of activating and repressing transcriptional factors localized to an ACH then determines its overall regulatory output, which dictates tissue-specific gene expression patterns[12–14]. Transcriptional factors, communication proteins, as well as non-coding RNA are suggested to be involved in the assembly of distant regulatory elements[11,15–18].

Gamma-herpesvirus replication takes place in specific tissues and viral gene expression occurs robustly in a highly-organized fashion. More than eighty viral genes are encoded by a relatively small viral genome, thus gene regulatory information such as the formation of ACHs is likely to be needed and embedded within the viral lifecycle as an accompaniment to DNA sequences required for efficient viral replication. Furthermore, such small viral episomes provide a unique experimental model to study 3D genetic interactions and their association with gene expression. Kaposi's sarcoma-associated herpesvirus (KSHV) episomes furnish a superb system for defining these mechanisms due to the combination of benefits derived from prior knowledge of defined viral transcriptional factors necessary to initiate viral gene expression, smaller genomic size, simpler genome organization, and a tractable genomic sequence using a bacterial artificial chromosome (BAC) system. Similar to cellular chromosomes, the viral genome is wrapped upon post-translationally modified histones and interacts with multiple histone modifying enzymes[19–21]. Such histone modifications correlate with transcriptionally "active", "poised", and "repressed" domains of viral chromatin in a manner equivalent to cellular chromosomes[22]. In addition, viral genomic loops were previously identified in KSHV-infected cells, in which CCCTC-binding factor (CTCF)-cohesin complexes establish physical looping of the KSHV genome[18]. Disruption of such genetic interactions by ablation of components of the cohesin complex deregulated the KSHV latency-lytic switch leading to viral reactivation[18,23]. These results

highlight the vital significance of higher-order viral genomic structure to the viral life cycle.

Like all herpesviruses, KSHV can exhibit two alternative life-cycles, known as latency and lytic replication. In latency, the viral genome persists in the host as low copy-number nuclear episomes, and its expression is largely silenced with the exception of several genes[24]. KSHV lytic replication is initiated by expression of a single viral protein, K-Rta. K-Rta is both essential and sufficient to induce lytic reactivation of the latent KSHV genome in the BCBL-1 cell line model, as well as in the de novo infection model[25].

In this study, we performed an unbiased chromosome conformation analysis by Capture Hi–C analysis, which is 3C combined with next-generation sequencing (NGS)[26]. We questioned if the KSHV genome is structured to maximize effects of K-Rta recruitment to achieve robust viral gene expression and also took advantage of an established reactivation system[27] to analyze spatiotemporal gene regulation in high resolution. We identified several KSHV genomic loops that encompassed K-Rta direct binding sites, and induction of K-Rta expression further induced genomic loop formation. These studies complement our recent report utilizing immuno-FISH approaches to study KSHV transcription in situ and examine KSHV gene expression in the 3D nuclear space of an infected cell[28]. The study showed that episomes responding to stimulation aggregated and recruited molecules that formed large "transcription factories". With the results reported here, we show that KSHV has evolved spatiotemporal gene regulatory mechanisms to assemble viral genomic domains and recruit RNA polymerase II, which results in effective viral gene expression from uniquely structured viral episomes.

## Results

**K-Rta binding sites on the KSHV genome**. The KSHV immediate early gene, K-Rta, is both necessary and sufficient to switch latent KSHV into the lytic infection cycle[27,29,30]. We speculated that robust gene activation by K-Rta associates with a unique structure formed by the KSHV genome, which is pre-fabricated with an architecture designed for K-Rta recruitment. To examine this hypothesis, the K-Rta binding sites in the KSHV genome were precisely mapped with ChIP-seq analysis in the context of the BCBL-1 cell line model. BCBL-1 is derived from a KSHV-infected human primary effusion (body cavity-based) lymphoma (PEL) and contains latent KSHV genomes. The TREx-(F3H3)-K-Rta BCBL-1 (hereafter called TREx-K-Rta BCBL-1) subline containing a Tet-inducible Flagx3 and HAx3-tagged K-Rta expression cassette was generated from TREx-BCBL-1 cells[27]. Doxycycline (Dox)-induced K-Rta expression (Fig. 1a, top panel), which consequently triggers KSHV reactivation, was confirmed by probing for the viral lytic protein K-bZIP (Fig. 1a, middle panel). The K-Rta recruitment sites were then examined by ChIP-seq analysis at 24 h post doxycycline treatment (Fig. 1b). The results showed multiple recruitment sites of K-Rta, including the promoter regions for ORF6, 7, 8, K2, 70, PAN RNA, 31, 45, K-Rta, K-bZIP, ORF57, K12, 73, and 75, in addition to both ori-Lyt loci (Supplementary Table 1). These results were in good agreement with previous promoter analyses that identified K-Rta responsive promoters in reporter assays conducted with 293 cells[31]. Among those promoters, the K-Rta response elements (RE) in the PAN and K12 promoters have been studied in detail[32,33], which allowed us to prepare mutant KSHV genomic episomes (described below).

**KSHV chromatin interactions**. Next, we examined the relationship between KSHV genomic structure and K-Rta

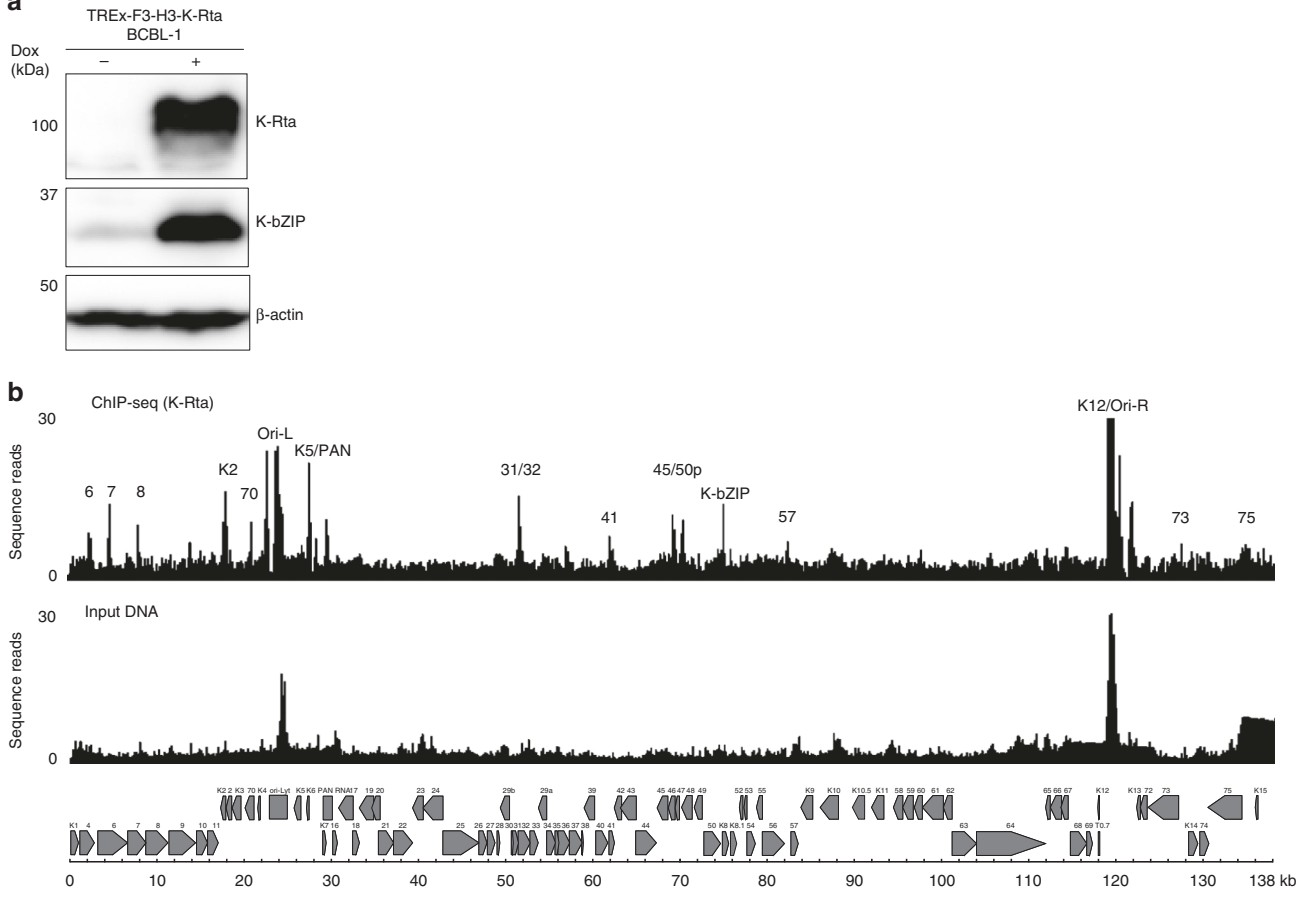

**Fig. 1** K-Rta recruitment sites on the KSHV genome KSHV reactivation. **a** KSHV lytic protein expression in TREx-K-Rta BCBL−1 cells during latency (−) and at 24 h post-dox treatment (+) was examined by immunoblotting. Blots were probed with the indicated antibodies. **b** K-Rta recruitment sites were examined by ChIP-sequencing analysis using K-Rta ChIP DNA prepared from TREx-K-Rta BCBL−1 cells 24 h post-dox treatment. Peak finding was performed with MACS2 program using the default parameter settings. Peaks mapped to the KSHV genome are plotted and the location of the each peak is indicated by the name of the nearest ORF name. Peaks are also summarized in Supplementary Table 1

recruitment sites before and during KSHV reactivation. The possible occurrence of long-range interactions was investigated by comprehensively examining all KSHV intra-genomic interactions using Capture Hi–C, a chromosome conformation capture (3C) approach coupled with target sequence capture and NGS[26]. The capture step was achieved using a custom library of biotinylated oligonucleotide baits (xGen Lockdown probes; IDT) that target the entire KSHV genome, including terminal repeats. This step was incorporated in order to enhance the sensitivity of Hi–C analysis to detect chromatin interactions of the KSHV genome by enabling substantial enrichment of KSHV-derived sequences (Supplementary Fig. 1). For this experiment, we used the same design as described above with TREx-K-Rta BCBL-1 cells and performed the capture Hi–C analysis from cells during latency and induced into the lytic cycle. We obtained 4656 (latent library) and 5520 (lytic library) sequence reads that were successfully ligated and mapped within the KSHV genome at the 29 *Bam*HI sites distributed along the ~ 150-kb KSHV episome. These ligation junction partners (*Bam*HI sites) were then visualized with Circos[34]. The number of sequence reads and identified genomic interactions are summarized in Table 1. KSHV genomic inter-actome maps depicting intra-genomic connections both before (latency) and after K-Rta induction (reactivation) are shown in Fig. 2a. The results showed that there is a trend for loop formation within similar functional genomic domains (topology domains), such as the latency cluster region and early-lytic gene

cluster region (Fig. 2a). The frequency of several of these latent interactions were highly significant above the empirical distribution of the population of detected ligation junctions ($p <$ 0.001) (Table 2 outlier test, part A). The relative genomic loop frequencies following reactivation are also depicted as a heatmap to visualize inducible genomic loop formation (Fig. 2b). When reactivation was triggered by induction of K-Rta expression, genomic loops among regions encoding early and immediate-early lytic genes, especially at the K-Rta promoter loci, were increased (Fig. 2b, *Bam*HI position #14). On the other hand, the number of genomic loops associated with the late and latent gene cluster regions (ORF63–ORF75; *Bam*HI sites #19–29 in Fig. 2a) did not increase significantly at this time point (24 h). In the presence of K-Rta, frequencies of genomic interactions were further increased between the PAN promoter (*Bam*HI position #5) and the K-Rta promoter locus (*Bam*HI position #14) (Fig. 2b and Supplementary Fig. 2), as well as additional loops engendered between the PAN promoter region and the downstream lytic gene cluster region. Genomic loops formed by contacts between the K2 promoter region (*Bam*HI position #2), which is another major K-Rta recruitment site (Fig. 1b), to the K-Rta promoter region (*Bam*HI, position #14), as well as to the PAN promoter region (*Bam*HI position #5) were also increased (Fig. 2b and Supplementary Fig. 2). Statistical analysis showed, with the exception of the K2/K-Rta promoter (#2/#14) interaction, that all of the highlighted induced pairs in Fig. 2b were confirmed to be highly

**Table 1 Capture Hi-C sequencing reads and genomic links of ligation products**

| BCBL-1 cells/iSLKr.219 cells | Total read pairs | Valid genome loops (Total) | Total genomic loops with KSHV DNA (host + viral) | Genomic loops between KSHV DNA fragments | KSHV-KSHV loops/total KSHV loops (%) |
|---|---|---|---|---|---|
| TREx K-Rta BCBL Dox(−) | 23,230,497 | 20,345 | 19,531 | 4656 | 23.8 |
| TREx K-Rta BCBL Dox(+) | 25,593,592 | 12,739 | 12,203 | 5520 | 45.2 |
| TREx K-Rta BCBL Dox (−)[a] | 51,479,787 | 920,102 | 656,969 | 122,956 | 18.7 |
| TREx K-Rta BCBL Dox(+)[a] | 49,486,847 | 977,802 | 698,569 | 236,110 | 33.8 |
| iSLK r.219 Dox (−)#1 | 30,807,748 | 58,035 | 51,779 | 5196 | 10.1 |
| iSLK r.219 Dox (+)#1 | 25,005,026 | 41,918 | 38,812 | 6224 | 16.0 |
| iSLK r.219 Dox (−)#2 | 24,713,145 | 13,799 | 9937 | 2569 | 25.9 |
| iSLK r.219 Dox (+)#2 | 24,903,367 | 15,414 | 11,215 | 3621 | 32.3 |
| iSLK r.219 Dox (−)#3 | 21,885,721 | 11,970 | 8857 | 2207 | 24.9 |
| iSLK r.219 Dox (+)#3 | 25,208,803 | 12,161 | 9056 | 3075 | 34.0 |

[a]Derived from *DpnII*. All other datasets are derived from *BamHI* ligation products.

significant above the background distribution of the data used to generate the heatmap ($p < 0.001$) (Table 2 outlier test, part B). Capture Hi–C interaction frequencies highlighted in Fig. 2b were also validated by qPCR (Fig. 2c).

**Inducible genomic loops and K-Rta recruitment sites**. To complement and confirm our Capture Hi–C results obtained with *Bam*HI digestions, we further generated a higher resolution genomic interaction map by performing Capture Hi–C analyses using the *Dpn*II restriction enzyme. The enzyme digests KSHV genomic DNA at 350 different sites and the frequent digestion allowed us to generate a linear genomic map by quantifying the Capture Hi–C sequencing reads encompassing the region at 3-kb intervals (Fig. 3). The linear heatmap of induced genomic looping formation was overlaid with the K-Rta ChIP-seq results (Fig. 1), which allowed visualization of the relationship between K-Rta recruitment sites and inducible genomic loop formation. A number of genomic interactions among early and immediate-early lytic encoding regions, especially near the K-Rta recruitment sites, were again elevated (Fig. 3 heatmap, red), and genomic links with late gene and latent gene clusters were either reduced or unchanged by K-Rta expression at 24 h post induction (Fig. 3 heatmap, blue). The results suggested that K-Rta binding sites are brought into proximity and aggregated in three-dimensional space via the formation of genomic loops during active transcription. Although only 2 of 13, 3-kb interval pairs that were evaluated attained statistical significance over the background distribution ($p < 0.001$) (Table 2 outlier test, part C), the overall frequency of genomic loops within the KSHV genome consistently increased during reactivation, while genomic links between KSHV and host chromosomes simultaneously decreased (Table 1). Similar loop frequency patterns were also observed in Capture Hi–C experiments using libraries prepared from recombinant KSHV-infected *i*SLK cells in three biological replicates (Table 1).

**Single K-Rta binding sites regulate distal gene expression**. The significance of K-Rta recruitment and its association with the genomic loops was next examined after preparing recombinant viral chromatin, which harbors two known K-Rta direct binding sites[32,33]. Using a two-step marker-less Red recombination approach with KSHV BAC16[35,36], we created a series of point mutations in either the PAN K-Rta response element (RE; PAN Mu), the K12 RE (K12 Mu), or both REs (double-knockout, DKO, Fig. 4a). A panel of *i*SLK cell lines infected with recombinant KSHV harboring K-Rta binding site mutations were subsequently established. Our mutations do not involve putative

CTCF binding sites, a factor associated with loop formation and which is known to recognize the CCA(C/G)(C/T)AG(A/G)(G/T) GGC core sequence[37]. Using the mutant episomes described above, K-Rta mediated viral gene expression was examined by RT-qPCR to define the importance of a single K-Rta direct binding site to the entire KSHV gene expression program. KSHV RT-qPCR arrays provided an expression read-out for every KSHV ORF over 72 h of reactivation (Fig. 4c and Supplementary Fig. 3). As expected, PAN RNA decreased dramatically in the PAN Mu cells. Surprisingly, disruption of the K-Rta-K12 binding site in the K12 Mu genome also diminished PAN expression, despite the presence of exogenous K-Rta and an intact K-Rta RE at the PAN promoter (Fig. 4c, inset). Similarly, K12 expression was reduced not only in K12 Mu cells but also in the PAN Mu cells (Fig. 4c, inset). As expected, the DKO virus was the most defective for expression of these two lytic genes (Fig. 4c and Supplementary Fig. 3). Interestingly, there were clusters of gene loci that were responsive to each particular K-Rta binding site mutation (Fig. 4d). The expression of a contiguous stretch of genes from K6-ORF43 was particularly sensitive in the PAN K-Rta RE mutation (Fig. 4d, upper and middle panels), ORFs marked in red and purple in the genome map); whereas for the K12 K-Rta RE mutation, the expression of several discrete clusters of genes that include ORF11-17, ORF37-40, ORF50-52, ORF54-55, and a latent gene cluster region were decreased (Fig. 4d, upper and lower panels), ORFs marked blue and purple in the genome map). Importantly, the affected ORF clusters encompassed genomic regions that are physically involved with the K-Rta-induced genomic looping events identified (Fig. 3), thereby providing validation of both the findings from the Capture Hi–C studies and their functional significance. In the *i*SLK cells, K-Rta is supplied exogenously in a dox-inducible manner, thus K-Rta expression is largely equivalent across the different recombinant KSHV-infected cell lines (Fig. 4b, c). However, endogenous K-Rta expression was also sensitive to both the PAN promoter–K-Rta interaction and the K12 promoter–K-Rta interaction (Fig. 4e). As expected, viral replication was diminished in mutant viruses, exhibiting reductions in encapsidated viral DNA copy number relative to BAC16 WT at 120 h post reactivation with Dox (Fig. 4f).

**Clustering of K-Rta binding sites at the KSHV ACH**. As capture Hi–C data (physical evidence) and KSHV PCR array results (functional evidence) indicated loop formation between regions of the KSHV genome containing the PAN RE and K12 RE, we asked whether binding of K-Rta at one RE would influence binding at the other, presumably proximal RE. K-Rta ChIP

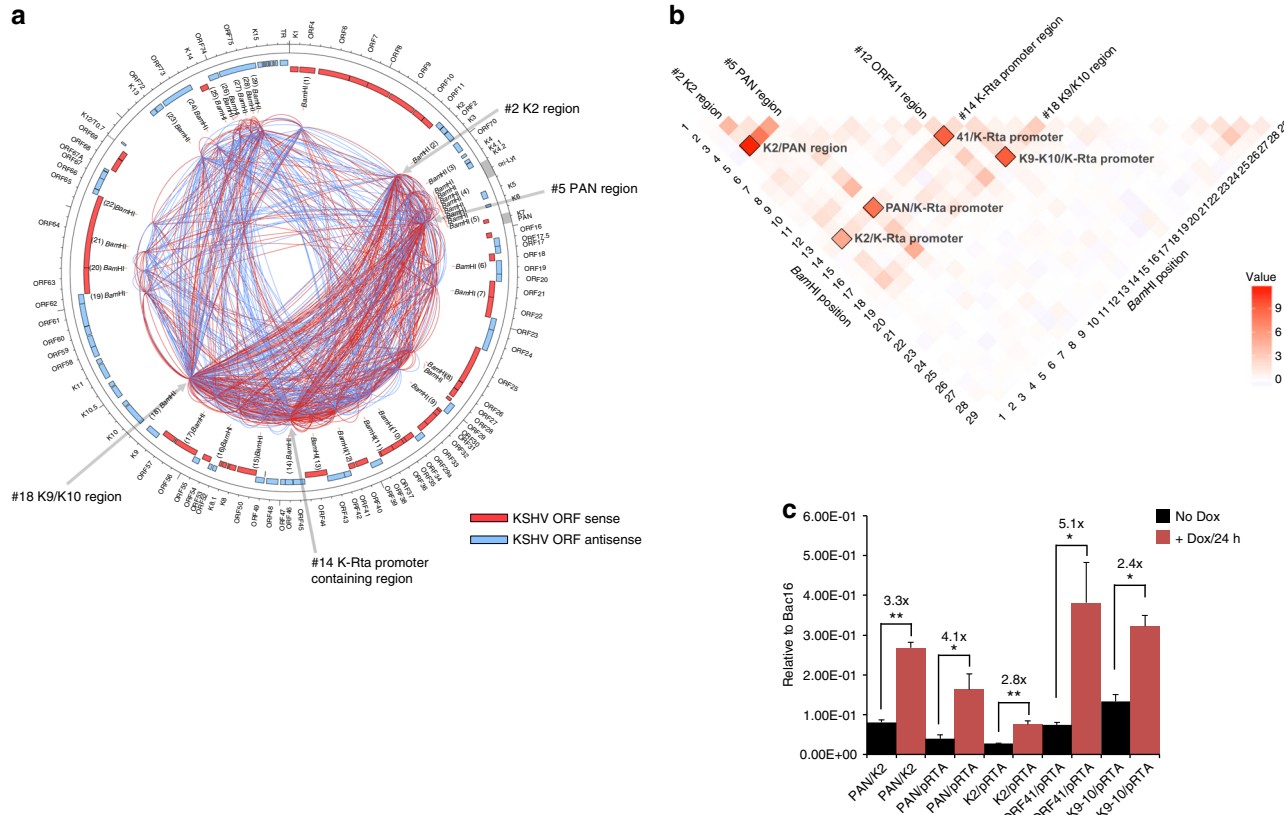

**Fig. 2** Comprehensive mapping of KSHV genomic loop formation in TREx-K-Rta BCBL-1 cells with Capture Hi–C. **a** Circos diagrams depicting KSHV genomic links detected by Capture Hi–C. Each arc connects two *Bam*HI fragments and represents a distinct interaction. Blue- and Red-colored lines indicate genomic links occurring before or after of K-Rta induction, respectively. Black outer hatches mark genomic positions. ORF names are indicated in the outer circle. Positions of non-coding RNA are shown by gray shading in the outer circle and position of *Bam*HI sites are also indicated. **b** Regulation of genomic loops by KSHV reactivation in TREx-K-Rta BCBL-1. The relative increase/decrease of genomic links by induction of K-Rta was obtained by subtracting the relative sequence reads of non-reactivated TREx-K-Rta BCBL-1 cells from those in reactivated TREx-K-Rta BCBL-1 cells, and visualized in a heatmap by using R package ggplot2. Values are sequence reads at each position divided by the total sequence reads contained within the KSHV genome and multiplied by $10^3$. The numbers indicate positions of the *Bam*HI sites, which correspond to the *Bam*HI site numbering shown with in the Circos plot in panel a. Several *Bam*HI sites and their corresponding column in the heatmap are labeled, and genomic loops that increased during reactivation are marked by black squares. *P*-values for selected loops are listed in Table 2 part A (constitutive) and part B (induced). **c** Enriched Capture-C DNA (*Bam*HI ligation products) was analyzed by qPCR for the contacts listed using DNA from TREx-K-Rta BCBL-1 cells±dox. Values (mean±SD, $n = 3$) represent signals obtained relative to a BAC16 *Bam*HI random ligation matrix. Fold increase (+dox/−dox) is listed. (*$p < 0.05$; **$p < 0.01$, Student's *t*-test)

analysis was performed using chromatin from dox-induced (0 and 8 h) iSLK BAC16 WT, PAN Mu, and K12 Mu cells and the degree of K-Rta recruitment to each RE was examined. In WT-infected cells, K-Rta binding was very robust at the K12 RE, but reduced binding at this RE was observed with either the K12 Mu or PAN Mu virus (Fig. 5a, right panel). A similar reduction in PAN RE binding was observed in cell lines containing either of the mutant viruses, although the overall binding of K-Rta was lower than that observed at the K12 RE (Fig. 5a, left panel). Taken together, these results suggest that the binding of K-Rta to its target promoters is dependent, at least in part, through clustering of the sites via genomic looping and the formation of ACHs.

**Ectopic PAN RNA does not rescue the PAN RE mutant**. One explanation for the global defect in gene expression for the K-Rta RE site mutant viruses is that each binding site mutation results in reduction in the amount of PAN RNA and/or K12 proteins produced and needed for efficient reactivation, rather than a genomic binding site/looping effect, per se. To address this, PAN Mu iSLK cells were first transfected with a PAN RNA expression vector 48 h prior to dox-induced reactivation. This raised the level

of PAN RNA 3000-fold above the level present in reactivated PAN Mu cells transfected with empty vector (Fig. 5b, left panel). Despite this level of PAN complementation, gene expression analysis demonstrated that the PAN Mu virus was still dramatically defective in lytic KSHV gene expression (Fig. 5b, right panel), indicating that activation of the PAN promoter and/or active transcription of PAN RNA is important for the transactivation of many other viral genes. These results are consistent with the fact that K12 Mu also impaired PAN RNA expression ~ 16-fold, yet the pattern of gene cluster expression is significantly different between the two mutants (Fig. 4d). If the presence of PAN RNA is the key, we should have expected to see similar gene expression profiles between the two mutants.

**K-Rta binding and inducible genomic loops**. Next, we examined effects of K-Rta binding on inducible genomic loop formation. Capture Hi–C analyses with iSLK/r.219 cells were first performed to probe differences of genomic loop formation by cell line, and we also generated a heatmap of inducible KSHV genomic loop formation in iSLK/r.219 cells (Fig. 6a, b). We have performed the Hi–C analyses with three biological replicates and confirmed

**Table 2 Statistical analysis of contacts**

| Contact pair[a] | Contact description | P-value |
|---|---|---|
| **A** | **Constitutive Loop Formation** | |
| **2/5** | **K2/PAN** | **<0.001** |
| 2/18 | K2/K9-10 | 0.871 |
| 5/14 | PAN/K-Rta promoter | 0.985 |
| **5/18** | **PAN/K9-10** | **<0.001** |
| 5/12 | PAN/ORF41 | 0.998 |
| 5/16 | PAN/K8.1 | >0.999 |
| **12/18** | **ORF41/K9-10** | **<0.001** |
| 14/12 | K-Rta promoter/ORF41 | >0.999 |
| **14/18** | **K-Rta promoter/K9-10** | **<0.001** |
| **18/23** | **K9-10/ORF72** | **<0.001** |
| 18/26 | K9-10/ORF74 | 0.985 |
| **B** | **Induced Loop Formation** | |
| **2/5** | **K2/PAN** | **<0.001** |
| 2/14 | K2/ K-Rta promoter | 0.8658 |
| **5/14** | **PAN/K-Rta promoter** | **<0.001** |
| **14/12** | **ORF46/ORF41** | **<0.001** |
| **14/18** | **ORF46/K9-10** | **<0.001** |
| **2/5** | **K2/PAN** | **<0.001** |
| **C** | **Induced Loop formation *DpnII* 3 kb interval** | |
| n/a[b] | 18001 to 21000 and 117001 to 120000 | >0.999 |
| n/a | **27001 to 30000 and 15001 to 18000** | **<0.001** |
| n/a | 27001 to 30000 and 18001 to 21000 | >0.999 |
| n/a | **27001 to 30000 and 21001 to 24000** | **<0.001** |
| n/a | 27001 to 30000 and 51001 to 54000 | >0.999 |
| n/a | 27001 to 30000 and 60001 to 63000 | 0.999 |
| n/a | 27001 to 30000 and 72001 to 75000 | >0.999 |
| n/a | 33001 to 36000 and 117001 to 120000 | >0.999 |
| n/a | 51001 to 54000 and 117001 to 120000 | >0.999 |
| n/a | 63001 to 66000 and 117001 to 120000 | >0.999 |
| n/a | 72001 to 75000 and 117001 to 120000 | >0.999 |
| n/a | 75001 to 78000 and 117001 to 120000 | >0.999 |
| n/a | 81001 to 84000 and 117001 to 120000 | >0.999 |
| **D** | **Induced Loop Formation *i*SLK/r.219 cells** | |
| 2/5 | K2/PAN | 0.091 |
| 2/14 | K2/K-Rta promoter | 0.352 |
| 5/14 | PAN/K-Rta promoter | 0.200 |
| 12/14 | ORF41/K-Rta promoter | 0.171 |
| 14/18 | K-Rta promoter /K9-10 | 0.588 |

[a]*Bam*HI site numbering; [b]not applicable. Statistically significant interactions are listed in bold
**A** Select potentially outlying pairs of positions were compared to the empirical distribution of data from the remaining pairs of positions $F_n(x)$, with P-values calculated based on extreme value theory as $1 - F_n(x)^n$. BCBL-1 TREx-F3-H3-K-RTA_BamHI_3C_No Dox Genomic loop numbers (Fig. 2a Circos plot, blue interactions) data were analyzed
**B** Potentially outlying pairs of positions (#2–#5, #2–#14, #5–#14, #14–#12, #14–#18) were compared to the empirical distribution of data from the remaining pairs of positions $F_n(x)$, with p-values calculated based on extreme value theory as $1 - F_n(x)^n$. The genomic loop data analyzed were that presented as the heatmap in Fig. 2b
**C** The data analyzed were that presented as the heatmap in Fig. 3. Statistical methods were the same as described for (A)
**D** Potentially outlying pairs of positions (#2–#5, #2–#14, #5–#14, #12–#14, #14–#18) were analyzed. The data analyzed were presented in the heatmap of Fig. 6b. Two sample *t*-tests performed on log-transformed differences were used to test if the change for each pair of positions of interest (2–5, 2–14, 5–14, 12–14, 14–18) differed from 0

dynamics of genomic looping formation to be very similar (Table 1). Representative figures from one of the triplicates are shown in Fig. 6. Similar to TREx-K-Rta BCBL-1 cells, genomic loops within lytic genes were increased compared to those of latent gene cluster regions (Fig. 6b, and Supplementary Fig. 4a). For example, genomic loops among PAN loci, Ori-Lyt, and K-Rta promoter loci were again elevated in the presence of K-Rta protein, indicating that the gene regulatory mechanism is conserved between the two cell lines (Fig. 6b, heatmap). The frequency of induced loops detected as significant in BCBL-1 TREx-K-Rta cells (Table 2, part B) did not attain statistical significance in *i*SLK/r.219 cells (Table 2, part D); this may be attributed to the lower proportion of fully reactivating cells in the population of the *i*SLK/r.219 cells. However, these Capture Hi–C results were indeed verified by qPCR analysis (Supplementary Fig. 4b). Next, we examined the relationship between K-Rta binding and induced genomic loops. Based on the Capture Hi–C and K-Rta

ChIP-seq results, we selected four *Bam*HI fragments that harbor K-Rta recruitment sites and contain regions where increased genomic looping formation with the PAN promoter region (#5 with bright red arrow) were detected in the presence of K-Rta (Fig. 6c red arrows). *Bam*HI fragments that do not contain a K-Rta binding site were also included as a comparison (Fig. 6c blue arrows). KSHV episome copy numbers in the *i*SLK cells were also measured by qPCR and found that PAN Mu contained 50% more KSHV episomes relative to WT in the *i*SLK cells (Fig. 6d). Effects of K-Rta binding at the PAN promoter on genomic loop formation was then examined with 3C analyses and the frequencies of induced genomic loops in PAN Mu and WT were compared. The results indicated that K-Rta recruitment to the PAN promoter is important for bringing other genomic loci into proximity of the PAN promoter region; however, *Bam*HI fragments that do not possess K-Rta binding sites were also brought into proximity of PAN promoter region in a K-Rta PAN RE dependent manner

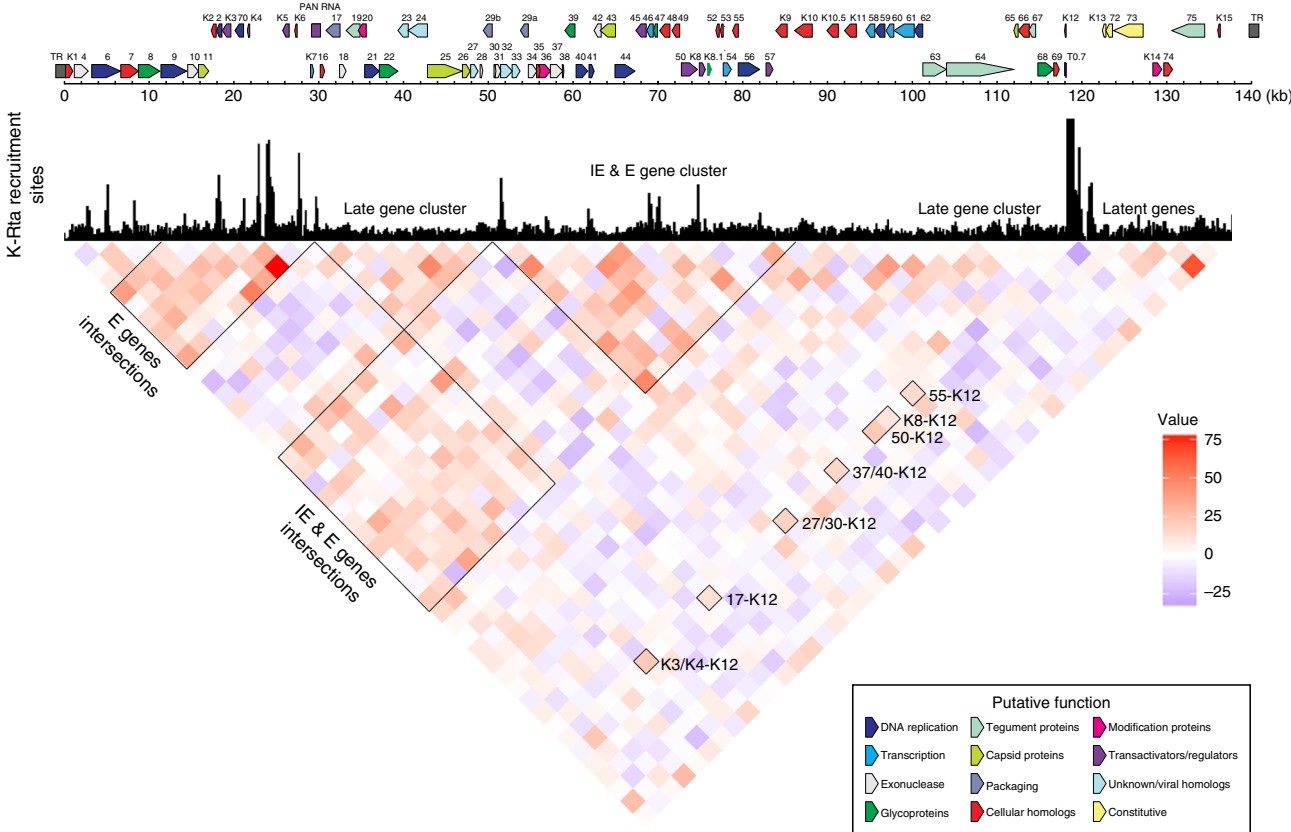

**Fig. 3** Dynamics of KSHV genomic loop formation. Total sequence reads in 3 kb intervals were counted and divided by the number of *Dpn*II sites within the 3 kb-sequence. Sequence read count per *Dpn*II site was then divided by total sequence reads containing KSHV genome to obtain relative number of genomic loops at the respective genomic region. Relative increase/decrease of genomic links by induction of K-Rta was then obtained by subtracting the relative sequence reads of non-reactivated TREx-K-Rta BCBL-1 cells from those of reactivated TREx-K-Rta BCBL-1 cells, and visualized by heatmap by using R-program. The ChIP-seq results presented in Fig. 1b is superimposed above the heatmap to visualize the relationship between loop dynamics and K-Rta recruitment sites. Genomic clusters representing IE, E, and late genes are marked. Loci where genomic loops that increased during reactivation are marked by block borders, and increased genomic loops with the fragment containing K12 K-Rta RE are also identified with relevance to the Fig. 4

(Fig. 6e, #5/#22). The results imply that activation of promoters is important for looping and that K-Rta is not required to be simultaneously bound to both fragments to form a genomic loop. Genomic loops formed between non-K-Rta enriched regions were also examined by qPCR (Fig. 6c, f, g. Two classes of contacts were observed in the non-K-Rta enriched loci: (i) contacts that appeared to be similar to K-Rta-enriched regions which were responsive to K-Rta induction, and were greatly reduced in PAN Mu cells (Fig. 6c, f), and (ii) non-K-Rta-enriched regions that were relatively non-responsive to K-Rta induction. This second type of interaction was detected at a similar level in both WT and PAN Mu cell lines (Fig. 6c, g), and showed no significant changes in gene expression at the genomic region by PAN Mu (Fig. 4d). Even though PAN-Mu *i*SLK/r.219 cells harbor 50% more KSHV episomes, the relative amount of pre-existing genomic loops in PAN Mu was lower than in WT, especially where gene expression by K-Rta induction was diminished (Figs. 4d and 6c). These results indicate that PAN promoter activation at the initial burst of transcription may be important to prepare latent chromatin structures for reactivation.

**Viral ACH and de novo infection.** KSHV reactivation initiated by supplementation of exogenous K-Rta in KSHV/Mu-infected *i*SLK cells allowed us to examine effects on specific viral promoter activation on the KSHV gene expression program. However, this masks the effects of initial activation of endogenous K-Rta expression by cellular factors and pre-deposited genomic loops

formed from direct binding sites. We have been searching for a suitable tissue culture system in which KSHV replicates to measurable degrees after de novo infection. To the end, we could demonstrate that lytic infection was robust in primary human gingival epithelial cells (HGEP) grown in trans-well culture with serum free media (Supplementary Fig. 5). Encapsidated viral genomic copy numbers were measured by qPCR and the DNA copy number was used to adjust for the de novo infection study. We infected primary HGEP cells with the maximal amounts of PAN Mu virus that was feasible to concentrate (M.O.I. of 50 viral copies/cell) and compared this with an equivalent amount of BAC16 WT virus. The results showed very weak eGFP signals in PAN Mu-infected cells (Fig. 7a). In contrast, cells receiving the WT virus were essentially ~ 100% positive for eGFP expression, and the signal intensity of infected cells was significantly brighter than that from PAN Mu-infected HGEP. KSHV gene expression array analysis further confirmed that viral gene expression was diminished with PAN promoter mutation, at levels which were from ~$10^4$ to $10^6$-fold less than WT BAC16 (Fig. 7b). Although it is currently not clear that the defect is due to lack of infectivity or impaired gene expression, the results clearly demonstrate the significance of activity of a viral ACH for the KSHV lytic replication.

## Discussion
Chromosomal loop formation between distal enhancers, silencers, and core promoters has emerged as a common feature among

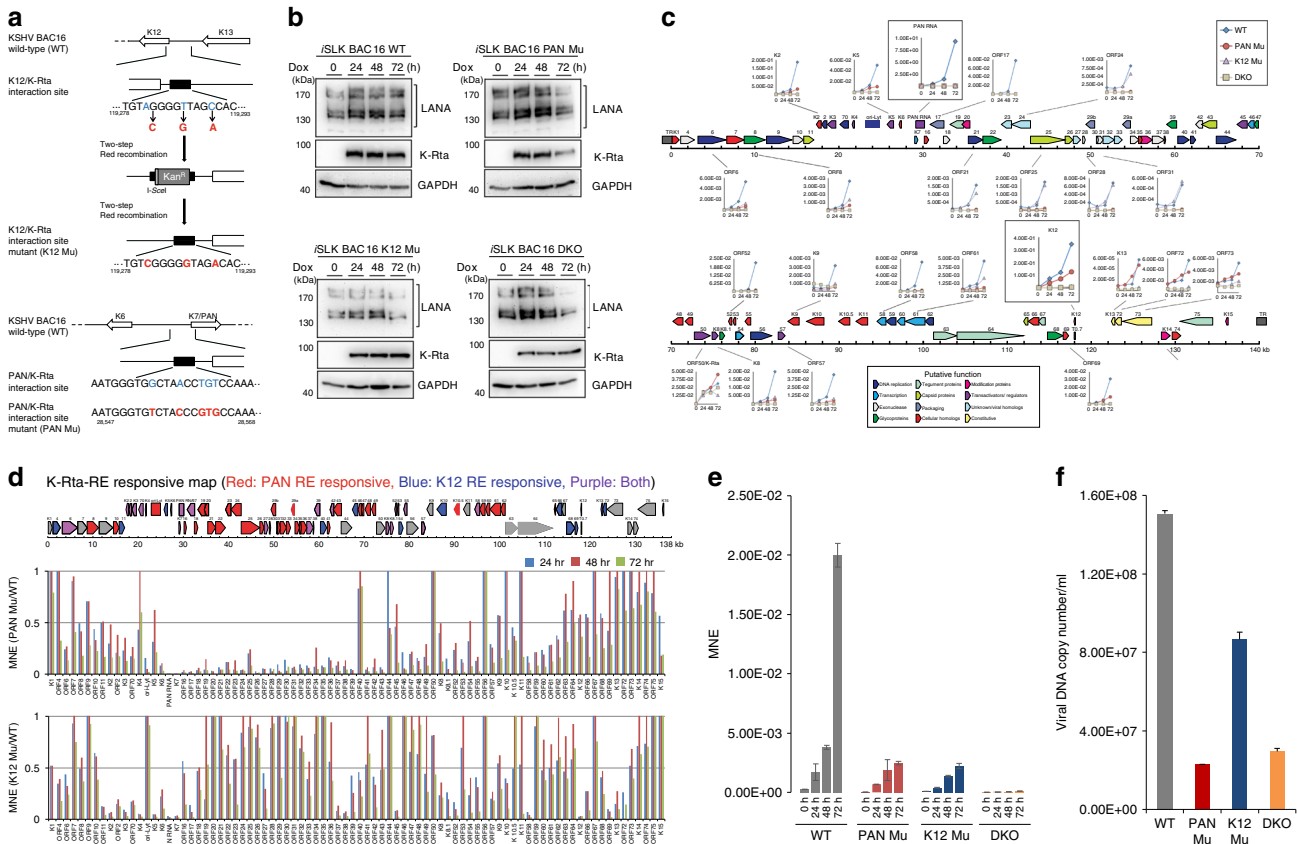

**Fig. 4** A K-Rta direct response element (RE) essential to coordinating regulation of KSHV gene clusters and reactivation. **a** Bacmid construction. Scheme of two-step red recombination for the construction of the KSHV K-Rta RE mutants used in this study: PAN Mu, K12 Mu, and PAN + K12 DKO. **b** Immunoblot analysis for KSHV gene products in *i*SLK BAC16 cell lines containing wild type and mutant KSHV bacmid clones. **c** Time course of KSHV gene expression in *i*SLK BAC16 cell lines. A map of the KSHV ORFs and their putative functions is shown. RT-qPCR expression analysis was performed for the indicated ORFs in WT (blue diamond) and PAN Mu (red circle), K12 Mu (purple triangle), and DKO (brown square) mutant lines. Values are mean normalized expression (MNE) using *ACTB* as reference. PAN RNA and K12 expression is presented as insets. Complete KSHV gene expression signatures are presented in Supplementary Fig. 3. **d** Mapping of genomic domains by viral gene expression. Normalized viral gene expression in each mutant KSHV (PAN Mu, K12 Mu) was compared with that of wild type at each time point to reveal dependency on K-Rta direct binding to the PAN RE (middle panel) and K12 RE (bottom panel). A gene exhibiting >50% reduction in expression at all time points (24 h, blue; 48 h, magenta; 72 h, green) during reactivation was considered as responsive. Values represent MNE relative to BAC16 WT (1 = unchanged). The upper panel summarizes genes regulated by PAN RE (red), K12 (blue), or both (purple). Genes unaffected by the effects of the mutations are marked in gray. Gene expression not evaluated was marked in black (ORF65). **e** Endogenous K-Rta gene expression. Endogenous K-Rta expression in *i*SLK BAC16 lines was measured using a gene-specific primer for reverse transcription, followed by qPCR. Values represent MNE±SD (*n* = 3). All 72 h means differ significantly (*p* < 0.01; one-way ANOVA with Tukey's post-hoc HSD test). **f** KSHV replication. Encapsidated KSHV DNA in culture supernatant 5 days post-reactivation was measured by qPCR. Values represent copy number/ml (mean±SD, *n* = 3). All means differ significantly in pairwise comparisons (*p* < 0.01; one-way ANOVA with Tukey's post-hoc HSD test), except PAN Mu vs. DKO

genomes of different organisms[38–40]. Our studies have implicated KSHV chromosomal looping as a mechanism to promote the KSHV latent-to-lytic switch. Previous studies by the Lieberman group have identified CTCF-cohesin complex-mediated KSHV genomic looping and its role in latency and reactivation[18,23,41,42]. We report here on new KSHV genomic contacts that may (a) prepare the latent genome for direct K-Rta binding events and (b) facilitate the formation of inducible genomic loops that are associated with distal promoter activation (Fig. 8).

The intricate nature of KSHV reactivation is exemplified by previous findings uncovered when this event is quantified on a single-cell basis; K-Rta-mediated reactivation appears highly stochastic with a variety of outcomes observed. Cells that are K-Rta positive undergo one of several fates, including complete reactivation, expression of a subset of delay-early genes, or abortive reactivation[43,44]. Only a small percentage of K-Rta-positive cells also express late proteins such as K8.1[44]. The studies

suggest that among cells harboring latent KSHV, a non-uniform population of viral episomes exist that may yield different outputs following the receipt of a reactivation signal. Our recent study confirmed this idea experimentally using RNA-FISH approaches to study KSHV transcription in situ, while examining KSHV gene expression in the 3D nuclear space of an infected cell[28]. A high degree of heterogeneity in the response of individual KSHV episomes to reactivation stimuli within a single cell was observed, and not all episomes in a reactivating cell express viral RNA. However, those episomes responding to stimulation appeared to recruit molecules that formed large "transcription factories"[28]. These outcomes are analogous to the emerging picture of eukaryotic gene expression when viewed on a single-cell basis; expression occurs in bursts, is inherently noisy, and is exemplified by highly stochastic expression levels (reviewed in refs. [45,46]). The underlying causes of this variation are still not entirely clear. Although our 3C results were obtained from cell populations,

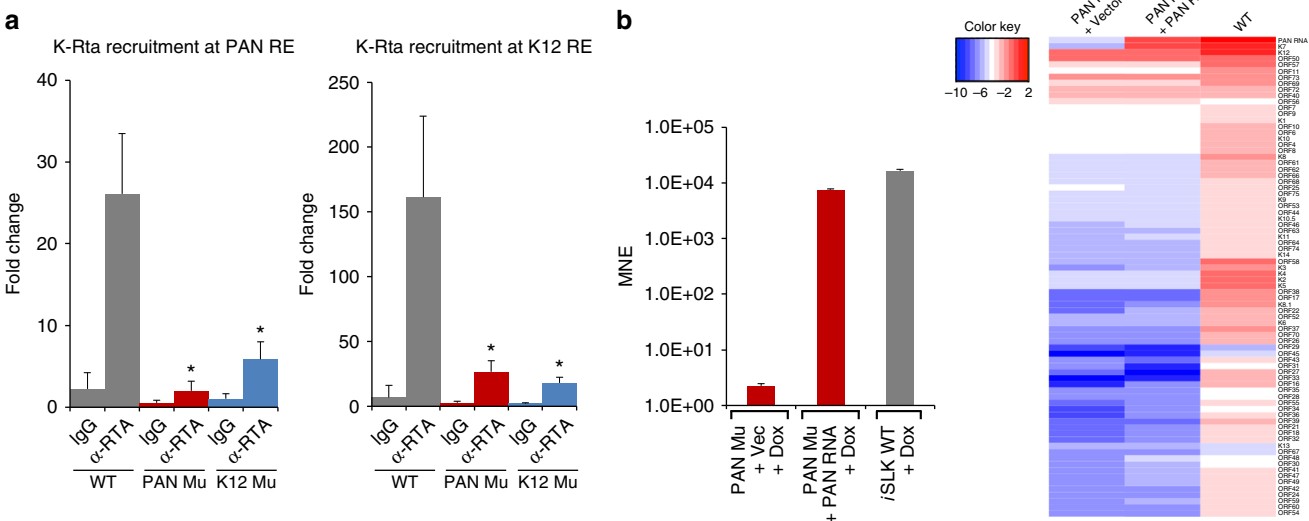

**Fig. 5** Lytic gene expression involves clustering of RE sites and requires PAN RNA in *cis*. **a** Cross-talk between the PAN RE and K12 K-Rta RE *i*SLK cell lines harboring BAC16WT, PAN Mu, or K12 Mu bacmids were left uninduced or induced with dox for 8 h. Cells were fixed and K-Rta ChIP was performed. Purified ChIP DNA was quantified by qPCR using primers that bracket the PAN RE (left panel) and the K12 RE (right panel) Values represent the mean±SD of three replicates. Fold change is shown relative to the ChIP/input DNA signal in the uninduced cells. One K-Rta RE mutation impaired recruitment of K-Rta to the other RE. All α-RTA means differ significantly in pairwise comparisons to WT ($p < 0.01$; one-way ANOVA with Tukey's post-hoc HSD test). Fold changes between PAN Mu vs. K12 Mu α-RTA means are non-significant. **b** PAN RNA complementation of PAN RE mutant does not rescue defective KSHV gene expression. (Left panel) PAN RNA expression. The expression of PAN RNA with/without complementation was quantitated by RT-qPCR and compared to the expression of BAC16 WT. MNEs are derived from RNA harvested 48 h after dox-induction. Values represent MNE±SD ($n = 3$). ACTB expression was used as the reference. (Right panel) KSHV PCR microarray. Individual viral gene expression was examined by KSHV PCR array and visualized as a heatmap. The heatmap was generated using Version 3.2.2 of R software with heatmap.3 package

individual episomes may vary in their looping architecture, as is the case with mammalian genomes[47], and loop diversity may contribute to variable reactivation scenarios described above. In this regard, we hypothesize that correct genomic KSHV loop formation and subsequent direct K-Rta binding events are additional factors to spur a subset of K-Rta-expressing cells and/or episomes towards full reactivation. Although the regulation of loop formation is unclear, examples in support of cells containing either preformed (hardwired) loops[48] or induced loop formation[49,50] have been reported. The loops we detected in KSHV-positive cells during latency suggest one class of KSHV genomic loops are hardwired. The observed induction of genomic links upon K-Rta expression also indicates the presence of inducible loop formation. Our study with mutant KSHV further suggests that activation of specific promoters, and presumably recruitment of RNA polymerase II complex, is important for inducible genomic loop formation (Fig. 6c). Although it is mostly speculative at this point, inducible genomic loops formed during the initial burst of lytic gene expression following de novo infection may dictate the 3D genomic structure as latency is subsequently established. This might explain why PAN Mu infected cells showed a lower number of genomic loops during latency, resulting in impaired gene expression during reactivation. It will be interesting to examine if manipulation of the viral gene expression program for lytic replication results in a different landscape of epigenetic marks during latency and leads to a different 3D genomic structure of latent chromatin.

In our studies, cells were fixed at 24 h post-reactivation in an attempt to minimize the contribution of viral DNA replication to contact formation. We also performed analyses using a 3C library prepared from cells reactivated in the presence of the viral DNA polymerase inhibitor, phosphonoacetic acid (PAA). The results showed there were little changes in the contact frequency of

several loci examined (Supplementary Fig. 6); this is also consistent with the fact that PAA usually does not down-modulate IE and E gene expression. Thus, the looping formation changes appear to be more than simply a reflection of genomic conformation changes induced by viral DNA replication, although a complete Hi–C analysis will be needed to fully resolve this question.

By performing capture Hi–C analyses with two different cell lines, TREx-K-Rta-BCBL-1 and *i*SLK/r.219 cells, we noticed that there is a difference in the hardwired loop formation between these two cell lines. A large number of genomic loops were found at the K9-K10 loci (*Bam*HI position #18) in BCBL-1 cells (Fig. 2a and Supplementary Fig. 2), but not in *i*SLK/r.219 cells (Fig. 6b). The KSHV K10.5 (LANA2) transcript has been shown to be expressed in KSHV-infected hematopoietic cells but not KS lesions[51]. Accordingly, KSHV may utilize additional ACHs in PEL cells to take advantage of transcriptionally active sites for effective viral reactivation, which may be established by active transcription at the corresponding genomic regions. We noticed that "poised" RNA polymerase II binding sites on the KSHV genome[52] significantly overlapped with the major K-Rta direct binding sites during reactivation. These results suggested that the KSHV genome is "poised" to be activated and stimulation of the K6-K7/PAN and/or K12/T0.7 regions through direct binding of the potent transactivator may lead to other viral promoter activation by releasing RNA polymerase II to traverse the KSHV genome.

To gain insight into how K-Rta binding to the KSHV genome leads to changes in chromosome conformation, we examined the relationship between genomic K-Rta binding sites (Fig. 1b) and previously published CTCF binding profiles[42]. The CTCF and cohesion complexes have been previously implicated in KSHV genomic looping[18,23]. Alignment of the profiles shows a partial overlap between CTCF and K-Rta enrichment peaks at regions encompassing PAN, K-Rta and K12 regulatory regions. This

result suggests that K-Rta is recruited to regions participating in genomic loop formation, and K-Rta may bring other enzymes to these critical regions and alter the KSHV genome structure (Supplementary Fig. 7a, b). Consistent with this, partial

knockdown (KD) of CTCF (Supplementary Fig. 8b) had a deleterious effect on the degree of K-Rta recruitment to ~ 90% of the peaks listed in Supplementary Table 1. Rad21 KD (Supplementary Fig. 8a) also had a weak, but opposing, effect on K-Rta

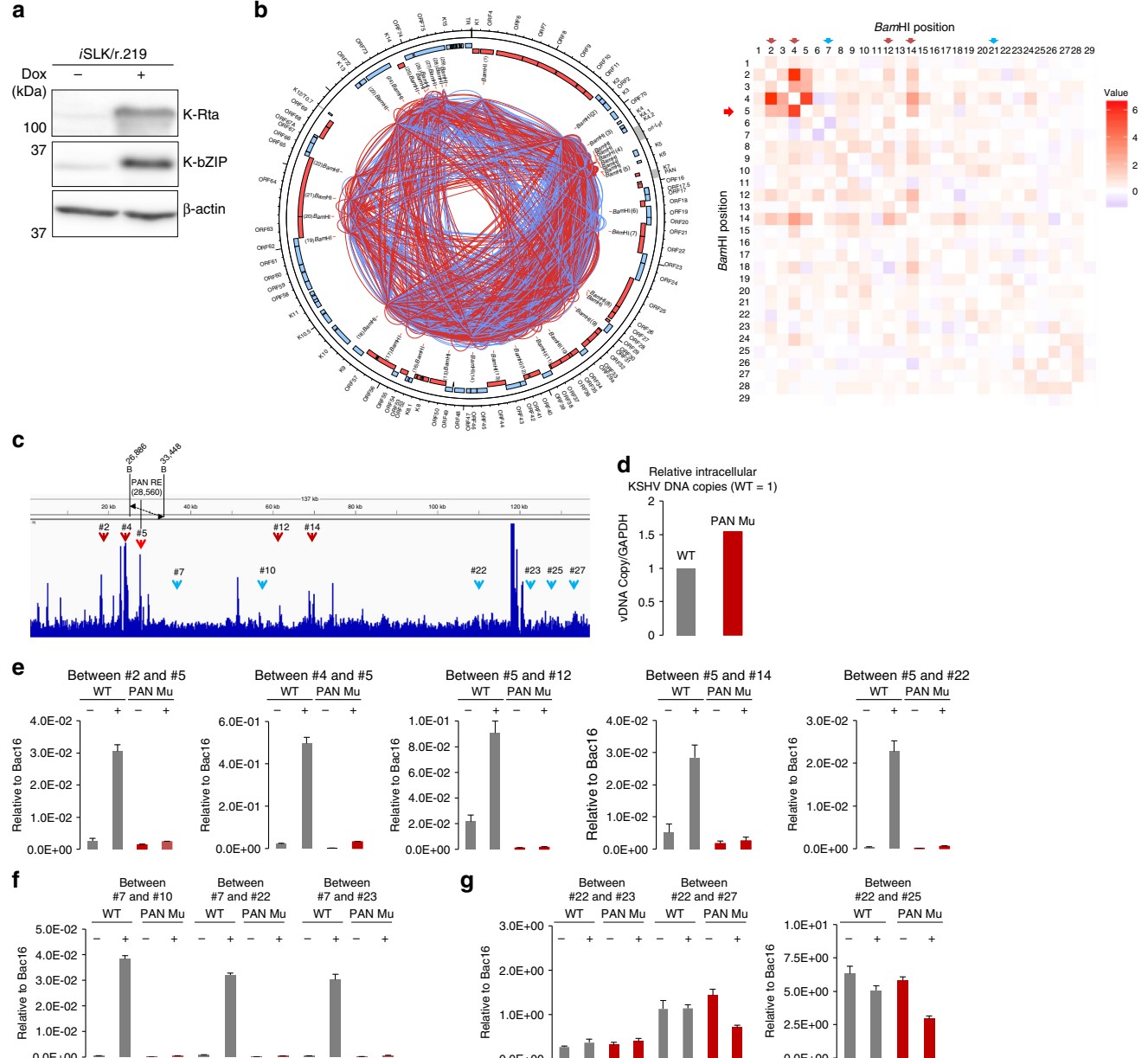

**Fig. 6** KSHV genomic loop formation in KSHV infected *i*SLK cells. **a** KSHV lytic protein expression. Immunoblotting analysis of K-Rta or K-bZIP expression in *i*SLK/r.219 cells during latency (−) and at 24 h post-dox treatment (+). **b** (Left panel) Circos diagram depicting KSHV genomic links detected by Capture Hi–C. Arcs indicate genomic links before (blue) or after (red) K-Rta induction in *i*SLK/r.219 cells. Black outer hatches mark genomic positions and ORF names are indicated in the outer circle. Position of *Bam*HI sites are indicated and numbered. (Right panel) Movement of genomic links during KSHV reactivation in *i*SLK cells. Increase/decrease of KSHV genomic loops was calculated similarly to that described in Fig. 2, and visualized by heatmap by using R package ggplot2. **c** Effect of K-Rta binding to PAN RE on loop formation. K-Rta ChIP seq plot and interaction sites are shown. The *Bam*HI fragment encompassing PAN is shown with anchor primers depicted. Target sites are indicated with red (K-Rta enriched) or blue (K-Rta valley) arrowheads. Arrowhead numbers refer to the *Bam*HI sites listed in the Circos plot in Fig. 6b. **d** Relative intracellular KSHV copy number WT = 1. Values were determined by qPCR on isolated total DNA from each latent cell line. **e** Real-time PCR analysis of 3C ligation products using anchor primers located on the *Bam*HI fragment containing PAN K-Rta RE (#5) with *Bam*HI acceptor primers as listed. PCR products were quantified relative to products from a BAC16 control *Bam*HI re-ligation matrix. qPCR results using a forward anchor/acceptor primer pair[18] are shown. Non-K-Rta enriched regions. **f** Induced contacts. Real-time PCR analysis of 3C ligation products (±dox 24 h) using an anchor primer located on the *Bam*HI fragment (#7) with *Bam*HI acceptor primers listed (#10, nt 36,700; #22, nt 111,299; #23, nt 123,970) and PCR products were quantified relative to products from a BAC16 control *Bam*HI re-ligation matrix. **g** Constitutive contacts. Real-time PCR analysis of 3C ligation products using an anchor primer located on the *Bam*HI fragment (#22) with *Bam*HI acceptor primers listed (#23, nt 123,970; #27 nt 134,049; #25 nt 129,537). Values (mean±SD, *n* = 3) represent signals obtained relative to a BAC16 *Bam*HI random ligation matrix

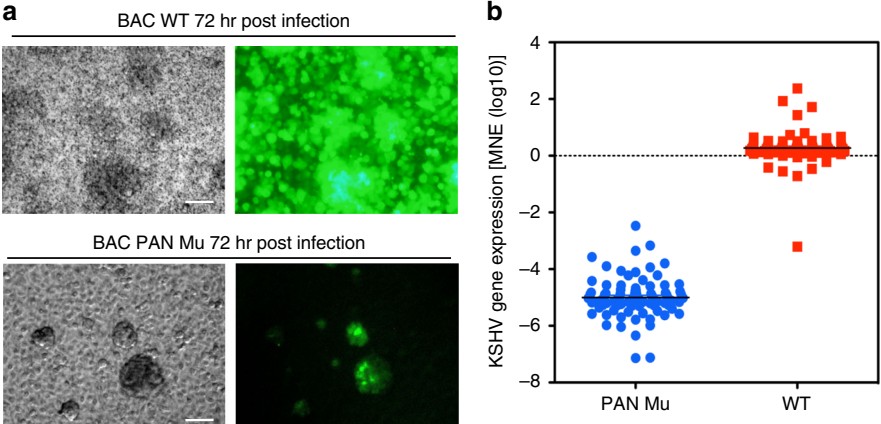

**Fig. 7** Effects on de novo infection. **a** Primary HGEP cells were cultured in trans-well and infected with concentrated viral supernatants from BAC16 WT or PAN Mu cultures at 50 viral DNA copies/cell. eGFP expression was captured 72 h post-infection. Phase contrast (left panels) and eGFP (right panels) are shown. Exposure time for image capture for eGFP was set to 3.11 s. Scale = 100 μm. **b** KSHV gene expression 72 h post-infection. KSHV gene expression was measured by KSHV PCR arrays and results were summarized as a box plot. Each point in the plot represents a KSHV ORF (PAN Mu, blue circles; WT, red squares)

binding, which exhibited modestly increased K-Rta occupancy at ~30% of the binding sites. A recent report has suggested that cleavage of Rad21 by cellular factors plays a role in KSHV chromosome conformational changes that accompany lytic reactivation, perhaps through opening of the cohesion ring[53]. Together, these results suggest that CTCF and genomic hubs prepared by the cohesin complex may have a role in K-Rta chromatin recognition. However, how K-Rta chromatin binding and/or activation of transcription regulates genomic looping requires further investigation.

In summary, we demonstrated here that the herpesvirus episomal genome and the latent-lytic infection switch represent a unique and effective experimental tool to study spatial and temporal gene regulation. Demonstration of KSHV replication in the HGEP cell model and the dynamic formation of viral ACH set a new stage for future KSHV gene regulatory studies. Identification of regulatory mechanisms of viral ACHs should lead to therapeutic intervention for KSHV-mediated malignancies.

## Methods

**Cells**. iSLK cells[54] were maintained in DMEM medium containing 10% fetal bovine serum (FBS; complete DMEM), 50 μg/ml G418, and 100 μg/ml hygromycin B in the presence of 5% $CO_2$. iSLK cells containing BAC16 WT or BAC16 mutants were cultured in complete DMEM containing 250 μg/ml G418, and 1000 μg/ml hygromycin and 1 μg/ml puromycin. TREx-(F3H3)-K-Rta BCBL-1 cells that expresses Flag × 3 and HA × 3 tags at the N-terminal region of K-Rta were generated and cultured in complete RPMI 1640 containing 50 μg/ml blasticidin and 100 μg/ml hygromycin B. Vero (ATCC CCL-81) and iVero (dox-inducible K-Rta expression) cells were cultured in the presence of 1 μg/ml puromycin. Human gingival epithelial cells (HGEP), obtained from pooled donors, were purchased from CellnTec (#HGEPp). The cells were cultured in CnT-prime epithelial cell medium (CellnTec #CnT-PR).

**Virus production**. iSLK cell lines containing each recombinant BAC16 were cultured in selection media in 150 mm culture dishes until 80% confluent. For reactivation, selection media was removed and cultures were re-fed with complete DMEM (w/o selection drugs) containing 1 mM sodium butyrate and 1 μg/ml doxycycline. The cultures were reactivated for 4 days. The cells were removed by centrifugation (180×g, 10 min) and the supernatants were passed through a 0.8-μm filter, and virus particles were concentrated by ultra-centrifugation at 25,000 rpm (112,000×g), for 2 h at 4 °C (Beckman SW28 rotor). Viruses were re-suspended in 500 μl of DMEM along with the residual ~500 μl media in the centrifuge tube and stored at −80 °C until use. Viral genomic copy number in the medium was measured by qPCR, as described previously[55].

**Plasmids**. To construct pENTR4-KSHV K-Rta/ORF50, K-Rta/ORF50 fragments were obtained by PCR and inserted into pENTR4 no ccDB (686-1) (Addgene plasmid # 17424)[56]. The PCR utilized pCINeo-3xFLAG-ORF50/RTA as template

and the primer pair sSalI-RTA(pENTR), CAT GTC GAC ATG GCG CAA GAT GAC AAG GG and asRTA-EcoRI(pENTR), CTG AGA ATT CTC AGT CTC GGA AGT AAT TAC GCC ATT G. To construct pCW57.1-KSHV K-Rta/ORF50, the KSHV K-Rta/ORF50 fragment was transferred from pENTR4-KSHV K-Rta/ORF50 to the Tet-ON lentiviral vector pCW57.1 (Addgene plasmid # 41393), by using the GATEWAY system (Invitrogen, CA, USA).

**Mutagenesis of KSHV BAC16**. The KSHV bacmid BAC16 was kindly provided by Dr. Jae U. Jung (University of Southern California) and mutagenesis performed according to previously described methods[35,36]. Point mutations were introduced at the K-Rta direct binding sites within the response elements of the PAN and K12 promoters[32,33].

The primer sequences for mutagenesis were as follows

(S-K12 RE Mut: ggtccacgctcacctctggcgcggccccgggaaatgggtgTctaCccccGacat aagcagtttgtcctacTAGGGATAACAGGGTAATCGATTT,

As-K12 RE Mut: cctgcgaaggggggcgtaaccgtaggacaaactgcttatgtCggggGtagAcacc catttcccggggccgcGCCAGTGTTACAACCAATTAACC

S-PAN RE Mut: tgttaatgacataaaggggcgtggcttccaaaaatgggtgTctaCccGTGccaaaat atgggaacactggTAGGGATAACAGGGTAATCGATTT

As-PAN RE Mut: caagctggcccctttttatctccagtgttcccatattttggCACggGtagAcaccca tttttggaagccacGCCAGTGTTACAACCAATTAACC,

Lowercase letters indicate identical sequence to KSHV BAC16, underlined uppercase indicates mutagenesis sites, and uppercase indicates the pEP-KanS sequence). Whole genome sequences of each BAC clone were confirmed by next-generation DNA sequencing as described below.

**KSHV BAC16 sequencing**. BAC16 DNA constructs were isolated from 2 l of LB culture with the Qiagen MAXI kit (Qiagen). Genomic DNA preparations were quantified using the Qubit Fluorometer (Life Technologies) and sheared to ~180 bp using the Covaris E220 Focused-ultrasonicator (Covaris, Inc.). Sequencing libraries were prepared using the NEBNext DNA Library Prep Kit (New England BioLabs, Inc.) and multiplexed sequenced (75 bp paired-end, 5 libraries/run) on the Illumina MiSeq platform (Illumina, Inc.) per manufacturer's standard protocols. Raw sequence reads (FASTQ format) were aligned back to the BAC16 KSHV reference (Human herpesvirus 8 strain JSC-1 clone BAC16, GQ994935.1) using BWA-mem[57] with default settings. Using the BAM alignment files as input, SAMtools mpileup[58] was then used to perform variant detection. Ultra-high coverage was obtained for most samples (~1200×) and allowed for comparative whole genome sequence analysis. The point mutations in each BAC16 were validated with unbiased variant calling (i.e., relative to the reference human herpesvirus 8 strain JSC-1 clone BAC16, GQ994935). The results also demonstrated a 24,710 C > T variant originating from the parental BAC16, and an additional mutation within the K12 promoter region of BAC16 K12Mu (position 119,316 A > G) (Supplementary Table 2).

**Generation of stable iVero cell lines**. The lentivirus vector was produced by tripartite-transfection of pCMV-VSV-G-RSV-Rev, pCAG-HIVgp and pCW57.1-KSHVK-Rta/ORF50 into 293 T cells (ATCC CRL-3216)[59,60]. The packaging plasmids (pCMV-VSV-G-RSV-Rev, pCAG-HIVgp) were a kind gift from and Dr. Hiroyuki Miyoshi (RIKEN, Japan). To establish Vero cells with inducible expression of K-Rta/ORF50 (iVero cell), Vero cells were infected with the lentiviral particles, and transduced cells were selected and maintained by culture in growth

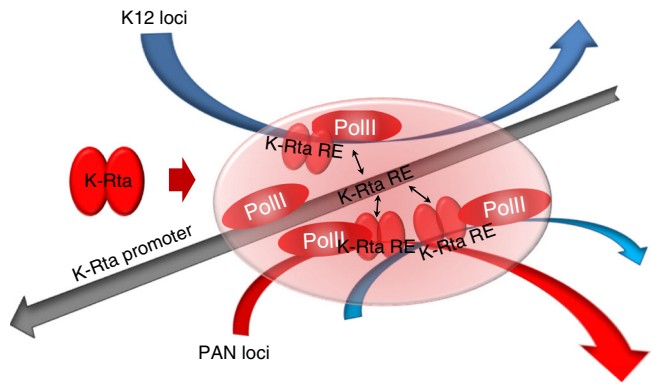

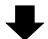

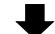

K-Rta expression further induces the formation of viral ACH
by clustering K-Rta RE

↓

Increases K-Rta & RNA polymerase II local concentration

↓

Efficient lytic gene expression

**Fig. 8** Proposed model of KSHV ACH. Multiple genomic regions containing K-Rta response elements (REs), including the locate the K-Rta promoter itself as well as other K-Rta responsive promoters, are positioned in close proximity to each other via 3D KSHV genomic structure. This mechanism facilitates K-Rta to activate promoters efficiently. Activation of "poised" RNA polymerase II at the K12 and PAN RNA loci then allows traveling to other viral promoters that are brought to in proximity during reactivation. Conversely, inhibition of a single ACH-associated promoter activation thus impairs multiple linearly distant viral gene expression

medium containing 2.5 μg/mL of puromycin (InvivoGen). KSHV BAC16 wild-type and mutants were transfected into iVero cells by the calcium-phosphate method, and stably transfected cells selected by culture in complete medium in the presence of 1,000 μg/mL of hygromycin B and 2.5 μg/mL of puromycin.

**Generation of stable iSLK cells**. The iSLK BAC16 lines were prepared by infection with viral supernatants produced in iVero cell lines described above. Each iVero BAC16 line was induced with 1 μg/ml doxycycline + 1 mM sodium butyrate for 4 days. Supernatants were collected, concentrated, and used to infect iSLK cells. For infection, iSLK cells were seeded into a 6-well plate ($2 \times 10^5$ cells/well), and viral supernatants were added to the cells on the following day. Two days later, the transduced cells were trypsinized and transferred to 10-cm dishes and cultured in complete DMEM containing 1 μg/ml puromycin, 250 μg/ml Geneticin and 1,200 μg/ml hygromycin B.

**Chromosome conformation capture**. The chromosome conformation capture (3C) procedure and analysis was carried out using chromatin derived from BAC16-WT or BAC-mutant infected iSLK cells. Latent or dox-induced iSLK cells were fixed with formaldehyde, digested with restriction enzyme (BamHI), diluted, and re-ligated. The primer design and protocol used has been described previously by Kang et al.[18]. A diagrammatic representation of the 3C PAN BamHI anchor primers are shown in Fig. 6c. Interaction frequencies between two KSHV fragments are based on the qPCR Ct values obtained using 3C chromatin template and a given anchor/acceptor primer pair relative to the value obtained using purified BAC16 DNA subjected to the 3C procedure.

**KSHV genomic capture probe library**. A custom KSHV genomic capture probe library using xGen Lockdown Probe technology (Integrated DNA Technologies, Inc.) was designed to target the entire KSHV genome sequence (human herpesvirus 8, NCBI RefSeq NC_009333.1) and manufactured by IDT. The library consists of a pool of 1,152 individually synthesized, 5′-biotinylated, 120mer oligonucleotide baits. A stock solution (0.75 μM) of the pool was prepared in TE (pH 8.0).

**Capture Hi–C analysis**. Long-range chromatin interactions within the KSHV genome were determined in an unbiased, comprehensive manner with Capture Hi–C, by utilizing 3C in conjunction with KSHV genome enrichment and next-generation sequencing (NGS)[26]. The chromatin prepared from KSHV-infected cells for the 3C procedure described above was used as input for Capture Hi–C. Briefly, infected TREx-K-Rta BCBL-1 or iSLK/r.219 cells ($1 \times 10^8$) were fixed with formaldehyde, and the chromatin was digested with either BamHI or DpnII restriction enzyme, and then

re-ligated under ultra-dilute conditions. DNA-protein crosslinks were reversed by incubation at 65 °C overnight with 0.2% SDS and the DNA purified with the QIA-quick PCR Purification Kit (Qiagen). DNA samples were submitted to the UC Davis Comprehensive Cancer Center Genomics Shared Resource for library preparation, target enrichment, and NGS. The DNA preparations were sheared to a mean fragment length of 250 bp using a Covaris E220 Focused-ultrasonicator (Covaris, Inc.) and sequencing libraries were then prepared from the DNA chimerae using the NEBNext DNA Library Prep Kit (New England BioLabs, Inc.). Target enrichment for KSHV genomic content was performed using a custom-designed KSHV genomic capture probe library (described above) and according to the manufacturer's standard protocols (IDT, Coralville, Iowa). Briefly, in-solution hybrid capture was performed by hybridizing libraries (500 ng) with the KSHV genomic capture probe pool (3 pmol) in a mixture containing xGen 1X Hybridization Buffer, Cot-1 (5 μg), and xGen Universal Blocking Oligos at a temperature of 65 °C for 4 h. Hybridized targets were then bound to streptavidin-coupled magnetic beads (Dynabeads M-270 Streptavidin beads; Thermo Fisher Scientific) by incubation at 65 °C for 45 min, and unbound DNA was removed by a series of high-stringency (65 °C) and low-stringency (room temperature) washes. KSHV genome-enriched Capture Hi–C library DNA was eluted and post-capture PCR enrichment performed with high-fidelity KAPA HiFi HotStart DNA Polymerase (Kapa Biosystems, Inc., Wilmington, MA) and purified with Agencourt AMPure XP magnetic beads. KSHV-enriched libraries were then validated with an Agilent 2100 Bioanalyzer, and quantitated with a Qubit fluorometer (Invitrogen) and by qPCR-based quantification (KAPA Library Quantification Kit). Libraries were submitted for sequencing (100-bp paired-end) on an Illumina HiSeq 2500 sequencing system. Junction sites (BamHI or DpnII sites) were mapped and visualized with Circos. Because of the relatively small size of the KSHV genome, exclusion of non-digested fragments with an additional capture step by the incorporation of biotin labeled nucleotide at the ends of successfully digested fragments was not performed. Consequently, we have had a relatively larger fraction of non-digested KSHV genomes that might be inaccessible for digestion by restriction enzyme, BamHI. In addition, there were a number of BamHI containing fragments that were unable to be mapped to BamHI sites within the short illumina sequence reads.

**Capture Hi–C NGS data analysis**. Paired-end sequence data (FASTQ format) was processed to identify putative Capture Hi–C interaction products (di-tags) using the Capture Hi–C User Pipeline (HiCUP) (Babraham Bioinformatics; http://www.bioinformatics.babraham.ac.uk/projects/hicup/). By executing a series of Perl scripts, this performed identification of the Hi–C ligation junctions within the reads (i.e., BamHI or DpnII sites), read alignment, and removal of PCR duplicates and common Capture Hi–C artifacts. Subsequently, read pairs derived from the interacting fragments were output in SAM format. Within HiCUP, Bowtie 2[61] was used to align FASTQ data to the reference KSHV genome sequence (Human herpesvirus 8 strain JSC-1 clone BAC16, GenBank: GQ994935.1) and an in silico BamHI-digested reference genome. Hi–C output was then analyzed with HOMER (Hypergeometric Optimization of Motif EnRichment; http://homer.salk.edu/homer/interactions/) to generate a Capture Hi–C tag directory and for identification of significant interactions. Junction sites (BamHI or DpnII sites) were mapped and visualized with Circos plotting (described below). Frequencies of chromatin looping sites are presented as the number of reads obtained with 3C of KSHV-infected chromatin.

**Circos plotting**. 3C sequence reads and genomic features of TREx-K-Rta BCBL-1 and iSLK r.219 cells infected with KSHV were visualized by using Circos plotting[34]. Circos software was selected for this analysis based on its ability to clearly display positional relationships among data derived from the KSHV genome. The Circos software program and companion tools were downloaded from its website (http://circos.ca.) Sequence reads were mapped to the KSHV reference genome (NC_009333.1). The number of sequence tags that mapped to a given BamHI or DpnII fragment is indicated in the Circos display and represent the number of unique interactions involving the fragment.

**Data visualization and interpretation**. The R package ver. 3.2.4 was used to visualize genomic links and gene expression. Heatmaps of genomic links were plotted with R package ggplot2 ver. 2.1.0. The gene expression profiles of iSLK-Wt and PAN-Mu cells presented in Fig. 5 were visualized with the R package Heatmap3 version 1.1.1. Those R packages were downloaded from the R Project website (http://www.R-project.org).

**De novo infection**. Human gingival epithelial precursor cells obtained from pooled donors were purchased from CellnTec. The cells were cultured in epithelial medium (CellnTec CnT-PR) as submerged cultures on polystyrene lifts in 12-well plates. Cells were bathed with 800 μl media in the upper chamber of the lift and with 1 ml medium in the lower well. When the cells were confluent, viral stocks were added to the upper chamber and the infection was allowed to proceed for 24 h. Cells were monitored for eGFP expression and RNA was purified using RNeasy columns (Qiagen).

**Chromatin immunoprecipitation assay**. Chromatin immunoprecipitation assay (ChIP) assays were performed, as described previously[62]. The antibodies used were rabbit anti-K-Rta IgG[63], mouse anti-K-Rta monoclonal antibody (a gift from Dr. Wood, University of Nebraska), anti-Flag M2 (Sigma F1804), and rabbit and mouse control IgGs (Cell Signaling Technology #2729 and #5415). Immunoprecipitated chromatin DNA was analyzed by SYBR Green-based quantitative PCR (qPCR) (Bio-Rad) with primers for each RE as follows: PAN RE-S (agcttgaaggat-gatgttaatg), PAN RE-As (gaagcggcagccaaggtgactg), K12 RE-S (ccaaga-gatccgtcctccgtgcc), and K12 RE-As (agcgggatgctaggtccacgc).

**ChIP-sequencing**. Briefly, chromatin DNA from $1 \times 10^8$ TREx- K-Rta BCBL-1 cells were used per assay and immunoprecipitated with 20 μg mouse anti-FLAG M2 (Sigma-Aldrich, F1804) or mouse control IgG (Santa Cruz Biotechnology, sc2025). 1 ng ChIP-enriched or input DNA was used to generate Illumina-compatible libraries with the KAPA LTP Library Preparation Kit (Kapa Biosystems, KR0453) according to the manufacturer's recommendations. Libraries were submitted for sequencing (50-bp single read) on an Illumina HiSeq 2500 sequencing system. The ChIP-Seq data was aligned to the human hg19 reference genome assembly and reference KSHV genome sequence (Human herpesvirus 8 strain JSC-1 clone BAC16, GenBank: GQ994935.1) with Bowtie 2[61]. Peak finding was performed with the MACS2 (Model-based Analysis of ChIP-Seq 2) program[64] according to the standard parameters described in the developer's manual. We used the default settings with a minimum FDR (q-value) cutoff of 0.05. The peaks and reads alignments were visualized using the Integrative Genomics Viewer (IGV) genome browser from the Broad Institute.

**Western blot analysis**. Cells were collected in RIPA buffer (50 mM Tris–HCl, pH 6.7, 1% NP-40, 0.25% sodium deoxycholate, 150 mM NaCl, 1 mM EDTA) supplemented with 1 mM PMSF and 1× protease inhibitor cocktail (Roche). Antibodies used for immunoblotting were anti-K-Rta[63], anti-LANA (Advanced Biotechnologies, Inc. #13-210-100, 1:1000) and anti-GAPDH (Millipore ABS16, 1:2000). The uncropped western blotting images are shown in Supplementary Fig. 9.

**RT-qPCR**. iSLK BAC16 WT and BAC16 mutant cells were reactivated with doxycycline (1 μg/ml) and RNA was isolated from these cells 0, 24, 48, and 72 h post-induction using RNeasy columns (Qiagen). cDNAs were synthesized using SuperScript III first-strand synthesis reagents (Invitrogen). Gene expression was analyzed by qPCR using specific primers designed by Fakhari and Dittmer[65]. KSHV gene expression was normalized to the cellular *ACTB* signal. RNA harvested from PAN RNA transfected cells included a DNase I treatment step.

**Data availability**. The data discussed in this publication have been deposited in NCBI's GEO Database under accession number GSE99950. The authors declare that all other data supporting the findings of this study are available within the article and its Supplementary Information files, or are available from the authors upon request.

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

## Acknowledgements

We are grateful to Dr. Kenichi Nakajima for assistance in HGEP tissue culture, Dr. Matthew L. Settles for advice in bioinformatics analyses, and Drs. Pei-Ching Chang, Jinjong Myoung, Charles Wood, and Jae U. Jung for providing reagents. We also thank Drs. Chie Izumiya, Feng Zhou and Mr. Christopher P. Chen for technical assistance. This research was supported by National Institutes of Health grants (DE025985) and by an American Cancer Society Research Scholar Grant (RSG-13-383-MPC). This work was also partially supported by grants from the U.S. Department of Agriculture (2015-67015-23268 and 2014-67015-21787). The UC Davis Comprehensive Cancer Center Genomics Shared Resource is supported by the NCI Cancer Center Support Grant (CCSG; NCI P30CA093373).

## Author contributions

M.C., C.G.T., and Y.I. designed research. M.C., T.W., K.N., R.R.D., Y.L., and Y.I. performed experiments. B.D.-J., M.C., K.N., Y.L., R.R.D., C.G.T., and Y.I. analyzed data. Y. L., R.D., and C.G.T. prepared sequence library and performed initial sequencing analyses. T.W. and M.F. prepared recombinant KSHVs and generated iVero cells. K.N. utilized the computer programs Circos and R to visualize Hi–C and qPCR data, respectively. M.C. and Y.I. wrote the manuscript and all authors edited the manuscript.

## Additional information

**Competing interests:** The authors declare no competing financial interests.

