## [Peer Review File · Nature Communications]

Reviewers' comments:

Reviewer #1 (Remarks to the Author):

The authors performed the Capture Hi-C analyses and found that association domains are induced by K-Rta expression. And these associations are attenuated by point mutations in the K-Rta responsive elements, resulting in the decreased expression of genes networked through the looping and the diminished KSHV replication. The authors provide multiple genomic and functional genetic studies to support their model. Overall, the findings are exciting and innovative, the work is technically impressive. However, there are several concerns described below that require additional experiments and further statistical analysis to substantiate the proposed model.

Specific Comments:

1. Reproducibility

Authors had conducted Capture-Hi-C with TReX K-Rta BCBL Dox -/+ cells two times using BamHI in fig2 and DpnII in fig3. But authors provided only one dataset in table2. All dataset used in manuscript should be provided in table2.

To compare the genomic loops in two cell lines, authors also had conducted Capture-Hi-C with iSLK/R.219 cells three times shown in table2.

Did authors used the combined result of three datasets in fig6? A better description of the methods used for statistical analysis of Hi-C is required.

2. Validation of Capture-Hi-C

a) The genomics data needs additional validation. The 3C analyses performed in iSLK/R.219 cells in fig6C-(c) should be provided with negative control loci. But there appears to be no differences between positive and negative association in fig6C-(c).

b) Also, 3C validation should be also conducted with TReX K-Rta BCBL cells since the majority of datasets (figure1~5) are acquired from TReX K-Rta BCBL cells and association pattern are quite different according to strains.

c)What type of correlation is there between the BamHI vs the DpnI based Capture 3C?

3. Significance of associations

There are limited numbers of BamHI sites. And the position of BamHI sites is biased. For example, #5 PAN region include 3~5 BamHI sites. How was the dataset processed to remove the background non-specific associations?

In fig2B, 3, 6B-(b), authors pointed out some significant association. It will be more reliable to use statistical analysis with p-value to call the significance.

4. Promoter-Promoter association

Authors mentioned about the association among promoters throughout the manuscript. But there is no evidence about that. Is it possible to provide any evidence to insist on the association among promoters by using high resolution of DpnII Capture-Hi-C data?

5. Statistical comparison between Capture-Hi-C and ChIP-seq data including CTCF/cohesin Authors suggested that genomic loops are formed among K-Rta binding sites based on the comparison of K-Rta ChIP-seq and Capture-Hi-C data. It will be more trustable to be tested statistically.

And I also recommend including the ChIP-seq data of CTCF/cohesin, which is already known as architectural proteins for KSHV genome, for the comparison.

6. The pattern of viral DNA copy numbers in pan mutant cells is quite different according to cells shown in fig4F and fig6C-(b). What is the possible hypothesis?

7. There is no analysis of existing loops prior to reactivation, and whether these are disrupted by reactivation? Are any loops lost during RTA-induced reactivation? Or only gained?

8. Is DNA replication required for 3C loop formation? Are the 3C interactions just a measure of the genome conformation change induced by lytic replication structure? The effect of viral lytic DNA replication inhibitors, such as PAA or gancyclovir, on 3C formation needs to be tested.

9. The assumption is that these 3C interactions are from the same genome, but potential clustering of multiple genomes may also account for these interactions, especially if lytic replication occurs at replication centers.

Minor

10. Discussion: line 391. It is not correct to say that the "late gene cluster", but rather the "latency gene cluster". Also, the "latency gene cluster" was not enriched in H3K27me3, but rather the "lytic gene cluster" is enriched in bivalent modification that includes PRC1/2 and H3K27me3.

Reviewer #2 (Remarks to the Author):

In this article Campbell et al. studied the KSHV genomic structure and its relationship with K-Rta recruitment sites with Capture Hi-C analyses. The authors identified several KSHV genomic loops that encompassed K-Rta direct binding sites and induction of K-Rta expression further induced genomic loop formation. They also observed that abrogation of K-Rta binding by point mutations in the K-Rta responsive elements, impaired the formation of inducible genomic loops, inhibited specific viral promoter activation, decreased the expression of genes networked through the looping. On the basis of their data author proposed that K-Rta mediated viral genomic architectural dynamics plays an essential role in herpesvirus gene expression and replication.

The paper is well written and of interest. The data are convincing and well presented.

However, the study lacks mechanistic data or input about how K-Rta binding leads to looping of the viral genome. One plausible explanation would be recruitment of CTCF-cohesin complexes. As author mentioned the role of CTCF-cohesin complexes in their introduction so they should at least check this possibility. Thus, although the study contains aspects of novelty and is certainly of interest the authors need to address the following concerns.

1) It is mentioned that CTCF-cohesin complexes establish physical looping of the KSHV genome. So it is important to look for changes in binding pattern of CTCF-cohesin complexes upon expression of K-Rta. Also is the binding site of K-Rta and CTCF-cohesin complexes in close proximity? This will shed some light on the mechanism of looping.

2) The authors indicate that ablation of components of the cohesion complex deregulated the KSHV latency lytic switch leading to viral reactivation. Although it is not essential but if possible it would be nice to see K-Rta binding on viral genome in this condition.

3) It is shown that merged interactome maps depicting intra-genomic connections both before and after K-Rta induction. It is very difficult to see the differences in interactome. The authors should show it individually.

5) Interactome maps in the mutants should be also shown individually. It is difficult to find changes in the merged images.

Dear Reviewers,

We would like to thank the reviewers for their insightful comments which have substantially improved the manuscript. We have addressed the reviewer's concerns by conducting additional experiments, adding new statistical analyses of our 3C data, modified the presentation of data, and made clarifications throughout the text. Your comments are italicized in Times New Roman and our responses are provided in Arial.

Reviewers' comments:

Reviewer #1 (Remarks to the Author):

The authors performed the Capture Hi-C analyses and found that association domains are induced by K-Rta expression. And these associations are attenuated by point mutations in the K-Rta responsive elements, resulting in the decreased expression of genes networked through the looping and the diminished KSHV replication. The authors provide multiple genomic and functional genetic studies to support their model. Overall, the findings are exciting and innovative, the work is technically impressive. However, there are several concerns described below that require additional experiments and further statistical analysis to substantiate the proposed model.

Specific Comments:

1. Reproducibility

Authors had conducted Capture-Hi-C with TREx K-Rta BCBL Dox +/- cells two times using BamHI in fig2 and DpnII in fig3. But authors provided only one dataset in table2. All dataset used in manuscript should be provided in table2.

We have included DpnII data sets in Table 1.

To compare the genomic loops in two cell lines, authors also had conducted Capture-Hi-C with iSLK/R.219 cells three times shown in table2.

Did authors used the combined result of three datasets in fig6? A better description of the methods used for statistical analysis of Hi-C is required.

The figure presented in fig 6 represents one of the three experiments. We have clarified in text (Page 12 line 270-274). We also included statistical analyses. Results of statistical analyses are summarized in new Table 2.

2. Validation of Capture-Hi-C

a) *The genomics data needs additional validation. The 3C analyses performed in iSLK/R.219 cells in fig6c-(c) should be provided with negative control loci. But there appears to be no differences between positive and negative association in fig6c-(c).*

We have added several control loci that were not enriched for K-RTA binding. We found two types of linkages among non-enriched sites. (1) Linkages responsive to K-Rta binding similar to enriched regions and (2) linkages that exhibited a high basal interaction frequency and that were only modestly responsive to K-RTA expression. The new data are in Figure 6c (d, e) and described in the text (page 13 line 295 - page 14 line 302)

b) *Also, 3C validation should be also conducted with TReX K-Rta BCBL cells since the majority of datasets (figure1~5) are acquired from TReX K-Rta BCBL cells and association pattern are quite different according to strains.*

We have used qPCR to validate several linkages highlighted in Figure 2b. The confirmatory results are in Figures 2c and S4b.

c) *What type of correlation is there between the BamHI vs the DpnI based Capture 3C?*

We utilized the *DpnII*-based Capture Hi-C to refine the interaction map to an interval with an increased density of 3 kb as depicted in Figure 3. Comparison of the induced genomic loops based on the *BamHI*-based map (Fig. 2b) against the *DpnII*-based map (Fig. 3) yields a similar overall interaction frequency landscape with induced loop formation concentrated in the early and immediate early gene clusters. We clarified the description in text (Page 9 lines 190 - 191).

3. Significance of associations

There are limited numbers of BamHI sites. And the position of BamHI sites is biased. For example, #5 PAN region include 3~5 BamHI sites. How was the dataset processed to remove the background non-specific associations?

This is an excellent point. Background, non-specific associations are removed very early in the analysis while the paired-end sequence dataset is processed with the Hi-C User Pipeline (HiCUP) (described in *Materials and Methods*). Specifically, HiCUP executes a series of Perl scripts, which accomplishes identification of the Hi-C ligation *bone fide* junctions within the reads (*i.e.*, *BamHI* or *DpnII* sites), read alignment, and removal of PCR duplicates. Importantly HiCUP (*i.e.*, “HiCUP Filter” script) also removes the common Capture Hi-C artifacts, such as “di-tags” where both reads map to the

same restriction fragment. To address your comment regarding the #5 PAN region, we offer the following response. To qualify as a valid genomic ligation, there has to be only one *Bam*HI in the sequence read. In addition, the *Bam*HI sites at position #5 generate 32-bp *Bam*HI fragments, and ligation of the resulting multiple small fragments were excluded as invalid sequence reads from the analyses. In addition, we have performed *Dpn*II digestion to further confirm the relative frequency of valid interactions (genomic loops) at the region.

In fig2b, 3, 6b-(b), authors pointed out some significant association. It will be more reliable to use statistical analysis with p-value to call the significance.

We have added Dr. Blythe Durbin-Johnson, PhD., a UCD School of Medicine statistician to our project. She has carried out statistical analysis to the data presented in Figures 2, 3 and 6. The statistical results are summarized in Table 2.

4. Promoter-Promoter association

Authors mentioned about the association among promoters throughout the manuscript. But there is no evidence about that. Is it possible to provide any evidence to insist on the association among promoters by using high resolution of DpnII Capture-Hi-C data?

We agree that interacting fragments may not always correspond to promoter/promoter or promoter/enhancer sequences. We have corrected the text throughout to avoid misleading.

5. Statistical comparison between Capture-Hi-C and ChIP-seq data including CTCF/cohesin
Authors suggested that genomic loops are formed among K-Rta binding sites based on the comparison of K-Rta ChIP-seq and Capture-Hi-C data. It will be more trustable to be tested statistically.

Great point. We have tried to identify statistical approaches to analyze the correlation between K-Rta binding sites (ChIP-seq) and inducible genomic looping regions (Hi-C) by collaborating with Dr. Matthew L. Settle, the director of UC Davis Bioinformatics core facility. Unfortunately, we could not find appropriate program or approaches to obtain statistical value for the correlation. However, we would like to note that we have generated point mutants of KSHV in Fig. 4 to specifically inhibit a K-Rta binding at specific sites and validated the significance of the inducible loops at the site in gene expression (Fig. 6).

And I also recommend including the ChIP-seq data of CTCF/cohesin, which is already known as architectural proteins for KSHV genome, for the comparison.

Great suggestion. We have included the ChIP-seq data of CTCF and found that these two proteins were recruited similar genomic region; however, their binding sites were not completely overlapped. The results may suggest that K-Rta targets the boundary of genomic loops. Figures were presented in Figure S7 and S8. Discussion was included in the text (Page 19 line 399 - 414)

6. The pattern of viral DNA copy numbers in pan mutant cells is quite different according to cells shown in fig4f and fig6c-(b). What is the possible hypothesis?

We are sorry for the confusion. The data presented in Figure 4f is the extracellular vDNA production, of which is reduced among the mutant lines following reactivation. However, the relative intracellular vDNA copy number during latency is presented in Figure 6c (b). This analysis was provided as a control to demonstrate similar intracellular vDNA amounts among the wild-type and PAN mutant lines (i.e equivalent amounts of episomes).

7. There is no analysis of existing loops prior to reactivation, and whether these are disrupted by reactivation? Are any loops lost during RTA-induced reactivation? Or only gained?

Loop formation during latency and following reactivation (24h) is presented in Figures 2a, 6b, and S2.

8. Is DNA replication required for 3C loop formation? Are the 3C interactions just a measure of the genome conformation change induced by lytic replication structure? The effect of viral lytic DNA replication inhibitors, such as PAA or gancyclovir, on 3C formation needs to be tested.

We have used qPCR to examine the effect of PAA on 3C linkage formation. We conducted an experiment in which PAA (0.5 mM) and dox (1 ug/ml) were added simultaneously to cultures of BCBL-1 TREx-F3H3-K-Rta and maintained for 24h. Using select 3C primer pairs we could not detect effect of PAA + dox on 3C product levels when compared to dox-induced cultures. This is also good agreement that PAA does not interfere with early-lytic gene expression in general. Control experiments demonstrated that this PAA treatment was non-toxic to latently infected cells. These new results are in agreement with the data we presented in this paper in which Hi-C data was derived from 24h dox-induced cultures. We had chosen this time point for our studies so as to harvest cells for analysis prior to the onset of viral DNA replication. However, a comprehensive Hi-C analysis will be needed to fully evaluate this question. The new data are presented in Figure S6. We included the data in supplemental Figure 6, and discussed in text (Page 18 lines 377-385).

9. The assumption is that these 3C interactions are from the same genome, but potential

clustering of multiple genomes may also account for these interactions, especially if lytic replication occurs at replication centers.

Great point. We also considered the point. Accordingly, we have attempted to detect linkages established in *trans* by using a single primer qPCR on 3C DNA templates. Only a product of a *trans*-interaction should be subject to amplification using a single primer which anneals to the same site on each identical and linked DNA molecule. We could not unambiguously detect these linkages, suggesting that the majority of interactions are in *cis* (intra-genomic).

Minor

10. Discussion: line 391. It is not correct to say that the “late gene cluster”, but rather the “latency gene cluster”. Also, the “latency gene cluster” was not enriched in H3K27me3, but rather the “lytic gene cluster” is enriched in bivalent modification that includes PRC1/2 and H3K27me3.

Thank you very much. We have corrected the text.

Reviewer #2 (Remarks to the Author):

In this article Campbell et al. studied the KSHV genomic structure and its relationship with K-Rta recruitment sites with Capture Hi-C analyses. The authors identified several KSHV genomic loops that encompassed K-Rta direct binding sites and induction of K-Rta expression further induced genomic loop formation. They also observed that abrogation of K-Rta binding by point mutations in the K-Rta responsive elements, impaired the formation of inducible genomic loops, inhibited specific viral promoter activation, decreased the expression of genes networked through the looping. On the basis of their data author proposed that K-Rta mediated viral genomic architectural dynamics plays an essential role in herpesvirus gene expression and replication.

The paper is well written and of interest. The data are convincing and well presented. However, the study lacks mechanistic data or input about how K-Rta binding leads to looping of the viral genome. One plausible explanation would be recruitment of CTCF-cohesin complexes. As author mentioned the role of CTCF-cohesin complexes in their introduction so they should at least check this possibility. Thus, although the study contains aspects of novelty and is certainly of interest the authors need to address the following concerns.

Thank you very much for support of our work.

1) It is mentioned that CTCF-cohesin complexes establish physical looping of the KSHV genome. So it is important to look for changes in binding pattern of CTCF-cohesin complexes upon expression of K-Rta. Also is the binding site of K-Rta and CTCF-cohesin complexes in close proximity? This will shed some light on the mechanism of looping.

We examined K-Rta and CTCF binding sites on KSHV genome. The results showed both transcriptional factors were recruited to similar genomic regions; however, two transcriptional factors recognize very close but different genomic regions. This analysis is presented in Figure S7. We have individually knocked down (KD) CTCF and Rad21 in BCBL-1 TREx-K-RTA cells for 3 days prior to dox-induced reactivation and then examined K-RTA binding (24h dox) to each of the 18 identified peak regions listed in Table S1. The KD of CTCF resulted in a uniform statistically significant decrease in RTA binding at 15/18 RTA enrichment regions when compared to control KD. These results suggest that either CTCF mediated loop formation may be needed for optimal RTA chromatin recognition and binding. These data are presented in Figure S8B and the results are mentioned in the discussion (Page 19, lines 399 - 414).

2) The authors indicate that ablation of components of the cohesion complex deregulated the KSHV latency lytic switch leading to viral reactivation. Although it is not essential but if possible it would be nice to see K-Rta binding on viral genome in this condition.

Partial KD of Rad21 prior to reactivation had a modest effect on RTA binding to its peak enrichment regions (6/18 peaks affected). The effect was weaker than CTCF KD detailed above. However, with Rad21 KD RTA binding was increased at 5 of the 6 loci that were altered. The data are presented in Figure S8a. The opposing effects of Rad21 KD versus CTCF KD on RTA binding await further study. We discussed this in text (Page 19 lines 407 - 414).

3) It is shown that merged interactome maps depicting intra-genomic connections both before and after K-Rta induction. It is very difficult to see the differences in interactome. The authors should show it individually.

We agreed. Individual maps are provided in Figure 2a.

5) Interactome maps in the mutants should be also shown individually. It is difficult to find changes in the merged images.

Agreed. We included individual maps in Figure S4a.

REVIEWERS' COMMENTS:

Reviewer #1 (Remarks to the Author):

The authors have responded adequately to the previous reviews.

The authors provide a wealth of data, and the experiments appear to be well conducted.

However, there remain a few minor concerns that would help to improve the manuscript.

1. The writing could be improved in some places to make the interpretation of the data more clear. For example: first sentence is awkward:
"The three-dimensional structure of chromatin organized by genomic loops engenders RNA polymerase II access distal promoters"

Please check throughout.

2. The Circos plots are still very difficult to interpret (and worse than the original version). It is not clear if these many interactions are those that are significantly enriched relative to control bacmid ligations. If not, then only interactions of significant enrichment relative to background bacmid ligations should be drawn. This may simplify the interaction network. The other graphical representations of interaction, such as that in Figs 2B, and 3, emphasizing only those interactions that are different from bacmid controls., seem much easier to interpret.

Reviewer #2 (Remarks to the Author):

The authors have addressed my comments. Manuscript is now suitable for publication in Nature Communications.

REVIEWERS' COMMENTS:

Reviewer #1 (Remarks to the Author):

The authors have responded adequately to the previous reviews.

The authors provide a wealth of data, and the experiments appear to be well conducted.

However, there remain a few minor concerns that would help to improve the manuscript.

1. The writing could be improved in some places to make the interpretation of the data more clear. For example: first sentence is awkward:

"The three-dimensional structure of chromatin organized by genomic loops engenders RNA polymerase II access distal promoters"

Please check throughout.

We have addressed writing issues throughout the text, including the first sentence mentioned above.

2. The Circos plots are still very difficult to interpret (and worse than the original version). It is not clear if these many interactions are those that are significantly enriched relative to control bacmid ligations. If not, then only interactions of significant enrichment relative to background bacmid ligations should be drawn. This may simplify the interaction network. The other graphical representations of interaction, such as that in Figs 2B, and 3, emphasizing only those interactions that are different from bacmid controls., seem much easier to interpret.

The original Circos plots have been re-inserted.

Reviewer #2 (Remarks to the Author):

The authors have addressed my comments. Manuscript is now suitable for publication in Nature Communications.

Thank you.